# Heterogeneous Customizable Personalized Federated Fine-Tuning Approach for Large Language Models

Xin Tong [* 1]  Baojiang Cui [1]

## Abstract

Personalized federated LoRA fine tuning has become a key approach to addressing data heterogeneity in distributed fine tuning of large language models (LLMs). Existing methods typically assume homogeneous personalization needs across clients, relying on dual LoRA or personalized calibration schemes. However, they fail to account for the heterogeneity of local personalization requirements and the conflicting optimization objectives in dual LoRA, limiting scalability and performance. To address this, we propose Het-CPFLoRA, a customizable heterogeneous federated LoRA fine tuning algorithm inspired by the decoupling properties of LoRA parameters. We employ a single adapter fine tuning scheme to mitigate conflicts between personalized and generalized optimization, decouple LoRA into generalized and personalized subspaces for local customization, and use SVD compression to integrate cross client generalized knowledge. During inference, we introduce an OOD oriented dynamic mechanism to adjust the weighting between personalized and generalized decoupling knowledge, improving performance on user data. Extensive experiments on two public benchmark datasets show that Het-CPFLoRA outperforms state of the art methods in both personalization and generalization across heterogeneous scenarios. The source code can be found at https://github.com/tongxin-tx/Het-CPFLoRA.

## 1. Introduction

The emergence of large language models (LLMs) has significantly advanced development across various domains.

[1]School of Cyberspace security, Beijing University of Posts and Telecommunications, Beijing, China. Correspondence to: Baojiang Cui <cuibj@bupt.edu.cn>.

*Proceedings of the $43^{rd}$ International Conference on Machine Learning*, Seoul, South Korea. PMLR 306, 2026. Copyright 2026 by the author(s).

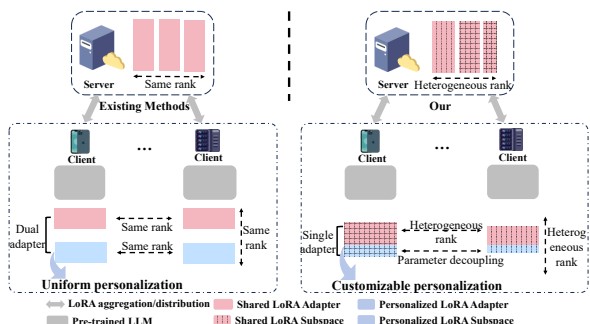

*(a)* Heterogeneous scalability comparison

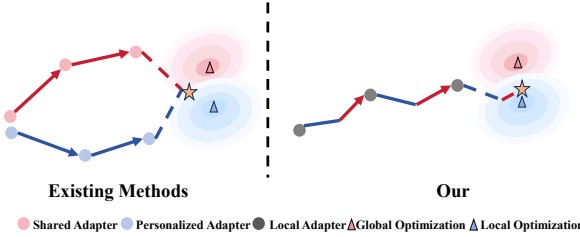

*(b)* Optimization direction comparison

*Figure 1.* Our motivation.

Fine-tuning these models with proprietary data enables them to perform exceptionally well across a wide range of tasks in fields such as healthcare, finance, and government, establishing a widely adopted learning paradigm (Parthasarathy et al., 2024). However, this paradigm requires substantial training data and computational resources to fully leverage the potential of LLMs. Consequently, it becomes unscalable for network nodes that are constrained by privacy concerns and limited computational capabilities (Ye et al., 2024a). Federated learning (FL) (McMahan et al., 2017) combined with low-rank adaptation (LoRA) (Hu et al., 2022) offers an effective solution by ensuring data privacy through collaborative fine-tuning and reducing computational overhead at edge nodes through LoRA training (Kuang et al., 2024).

However, data across different edge nodes exhibits heterogeneity (Ma et al., 2022). For instance, some nodes focus on domain specific question answering tasks, while others are dedicated to reading comprehension. This heterogeneity makes it challenging for FL to learn a single global model that performs well across all tasks. To address this challenge,

personalized federated learning (PFL) (Tan et al., 2022) has emerged, focusing on designing personalized models for each client while learning generalized knowledge. Currently, extending PFL to LLMs to leverage their rich pretrained knowledge for fine-tuning heterogeneous downstream tasks is a research focus. Existing approaches (Long et al., 2024; Hao et al., 2025; Peng et al., 2025; Shen et al., 2025) almost achieve personalized learning by alternating between generalized and personalized multi-LoRA adapters or by designing personalized calibration schemes.

These works face two challenges. First, the alternating learning across multiple adapters amplifies inconsistencies in optimization objectives. Specifically, the generalization adapter focuses on learning universal knowledge, while the personalization adapter targets personalized knowledge. Iterative optimization often exacerbates the conflict between these goals, leading to noise and suboptimal solutions during parameter merging, thereby suppressing model performance (Fig.1b). Second, it only allows for uniform personalized learning settings, failing to allow for heterogeneous personalized requirements (Fig.1a). Different clients exhibit heterogeneous demands for personalized and generalized knowledge learning. For instance, in clients requiring high expertise (e.g., medical or financial domains), uniform personalized settings may suppress the development of personalized learning representations for local tasks. For clients with high demands for generalized knowledge (e.g., education domain), excessive personalized learning may lead to overfitting and even undermine the transfer of shared knowledge across different clients. Therefore, *Our motivation is to design a fine tuning paradigm that allows clients to customize local personalized and general knowledge while mitigating conflicts between personalization and generalization optimization objectives.*

Driven by this motivation, we propose Het-CPFLoRA, a customizable, heterogeneous personalized federated LoRA fine tuning algorithm. It utilizes the LoRA matrix decoupling property we discovered (see Section 4.1), reducing multi adapter fine tuning to single adapter fine tuning to suppress optimization conflicts (Fig.1b) and achieve flexible, personalized customization of local heterogeneous models (Fig.1a). Specifically, it consists of three key steps: (1) Local Heterogeneous Training. Based on local LoRA configurations and personalized settings, it configures LoRA modules of varying sizes for each client, decouples the selection of personalized and shared subspaces, and initiates local training. (2) Global Heterogeneous Aggregation. It performs full rank aggregation on the heterogeneous shared subspaces (generalized subspaces) of each client and applies SVD based heterogeneous decomposition. (3) Dynamic Weighted Inference. It dynamically assigns weights to personalized and shared subspaces to improve response quality on unseen data. Our contributions are summarized as follows:

- We propose Het-CPFLoRA, the first federated LLMs heterogeneous personalization fine-tuning scheme that explores locally personalized level customization. It scales to local environments with heterogeneous data, computational resources, and personalization requirements, handling various client specific tasks.

- We reveal the unique decoupling properties of LoRA parameters, enabling flexible customization of client-side heterogeneous models. Based on this, we also introduce dynamic weighted inference using locally decoupled parameters, guided by user data out-of-distribution (OOD) patterns, which improves the quality of user data inference results.

- We apply Het-CPFLoRA to two datasets containing 13 downstream tasks for fine-tuning and inference on the LLaMA model. Experimental results show that Het-CPFLoRA outperforms state-of-the-art benchmark methods in heterogeneous, half heterogeneous, and all heterogeneous environments, achieving up to 1.91% improvement in generalization capability and up to 13.09% enhancement in personalization capability.

## 2. Related Work

### 2.1. Federated LLMs Fine-Tuning

In privacy constrained scenarios, federated fine tuning of LLMs leverages their extensive pretrained knowledge to perform complex downstream tasks while ensuring data privacy. Parameter efficient fine tuning (PEFT) methods (Xu et al., 2023), such as Prompt Tuning (Liu et al., 2022) and LoRA (Hu et al., 2022), are commonly used in LLM federated fine tuning for their ability to reduce training costs. FedPepTAO (Che et al., 2023) addresses client drift through adaptive prompt optimization, while FedBPT (Sun et al., 2024) introduces a black box, gradient free approach for federated prompt optimization, reducing variable exchange and enhancing communication efficiency. FIT (Ye et al., 2024b) utilizes LoRA for federated fine tuning across multiple tasks by averaging the $B$ and $A$ matrices. FLoRA (Bai et al., 2024) and FlexLoRA (Wang et al., 2024a) extend LoRA fine tuning to heterogeneous computing environments, using stacking and singular value decomposition methods, respectively. As LoRA incurs no additional inference overhead, it is widely adopted in federated LLM fine tuning tasks, and our work builds upon this approach.

### 2.2. Personalized Federated LLMs Learning

Most existing approaches utilize LoRA adapters to achieve personalized federated learning for large language models. FedDPA (Long et al., 2024) iteratively trains global and personalized adapters to learn both generalized and personalized knowledge. PF2LoRA (Hao et al., 2025), similar to

FedDPA, achieves personalized fine-tuning through a two-stage learning process involving a generalized adapter and a personalized adapter. What sets it apart is its ability to adaptively determine the appropriate LoRA rank. These works share a common feature: they employ multiple adapters to separately learn generalization and personalization, but overlook the potential optimization conflicts arising from this separate iterative learning. Furthermore, maintaining multiple adapters increases both local fine tuning and communication overhead.

pFedSeq (Peng et al., 2025) and pFedGPT (Shen et al., 2025) are recent personalized learning methods that differ from the multi adapter approaches described earlier. pFedSeq uses a learnable hypernetwork to process client adapter update sequences and generate personalized calibrations to adjust global adapters. pFedGPT applies hierarchical bayesian optimization to search for optimal module specific weights, enabling a nuanced integration of global and local LoRA to meet personalized learning. However, maintaining the hypernetwork and implementing hierarchical bayesian optimization pose significant overhead challenges.

Importantly, none of the above research addresses highly heterogeneous scenarios, such as varying client resources and local personalization needs. In contrast, our approach leverages LoRA's parameter decoupling to reduce multiple adapters to a single one, aligning optimization objectives. It also innovatively bridges the gap in PFL for heterogeneous scenarios and can be easily extended to applications with heterogeneous personalization and computing resources.

## 3. Preliminaries

### 3.1. Low-Rank Adaptation

LoRA is a PEFT method that injects low rank trainable matrix modules into specified layers of the base model. During fine tuning, it trains only these two low rank matrices $B \in \mathbb{R}^{d \times r}$ and $A \in \mathbb{R}^{r \times k}$ to approximate fully parameterized updates. $d$ and $k$ denote matrix dimensions, and $r$ denotes the rank. The updated model weights are $W = W_0 + \Delta W = W_0 + BA$. $W_0 \in \mathbb{R}^{d \times k}$ represents the original parameters of the pretrained base model, $W \in \mathbb{R}^{d \times k}$ represents the fine tuned model parameters, and $\Delta W \in \mathbb{R}^{d \times k}$ is the LoRA updated parameters. This approach greatly reduces the number of training parameters, thereby lowering computational resource consumption.

### 3.2. Federated Fine-Tuning Based on LoRA

Federated fine tuning using LoRA adapters has emerged as a popular approach for training LLMs. It primarily involves training LoRA adapters locally and sharing them with servers to enable fine-tuning of LLMs. Its objective function is defined as Eq.1, aiming to train an optimal global

adapter. Considering the heterogeneity of client-side local data, the personalized federated LoRA fine-tuning modifier objective function (Eq.2) addresses this challenge by training the locally optimal adapter for each client.

$$\min_{\Delta W_g} \mathcal{L} = \frac{1}{C} \sum_{c=1}^{C} \mathbb{E}_{X \sim D_c} \mathcal{L}_c(X|W_0, \Delta W_g) \quad (1)$$

$$\min_{\Delta W_1, \ldots, \Delta W_C} \mathcal{L} = \frac{1}{C} \sum_{c=1}^{C} \mathbb{E}_{X \sim D_c} \mathcal{L}_c(X|W_0, \Delta W_c) \quad (2)$$

where $C$ is the total number of clients, $D_c$ is the local dataset of client $c$ and $L_c$ is the expected local loss. $\Delta W_g$ and $\Delta W_c$ represent the global and local LoRA parameters.

## 4. Method

### 4.1. LoRA Matrix Decoupling

We discovery that LoRA update matrix parameters exhibit decoupling properties (Eq.3). Leveraging this, local knowledge can be divided into shared and personalized knowledge. Furthermore, clients can control the amount of heterogeneous generalized knowledge to be shared and the personalized knowledge to be retained. To illustrate this decoupling, consider fine-tuned matrices $B \in \mathbb{R}^{3 \times 3}$ and $A \in \mathbb{R}^{3 \times 3}$, the updated knowledge of the model is $BA \in \mathbb{R}^{3 \times 3}$. If the $B_s A_s$ and $B_p A_p$ matrices are obtained by splitting the fixed columns of the $B$ matrix and the fixed rows of the $A$ matrix, the sum of the $B_s A_s$ and $B_p A_p$ matrices is equivalent to $BA$. This matrix decoupling property offers another point of view for enabling federated personalized LoRA. Clients can customize shared and personalized knowledge, dividing the $B$ and $A$ matrices into shared ($B_s A_s$) and personalized zones ($B_p A_p$). $BA$ can be represented as the coupling of shared knowledge and personalized knowledge.

With regard to decoupling strategies, it is worth noting that the partitioning of $B$ and $A$ must be valid, i.e. rows must correspond to columns. In other words, the partitioning of columns $i$ to $j$ of matrix $B$ and rows $i$ to $j$ of matrix $A$ is used as the personalized or shared subspace.

### 4.2. Het-CPFLoRA Framework

Het-CPFLoRA is a heterogeneous PFL approach that enables the customization of both generalized knowledge and personalized knowledge learning for each client. The overall framework is shown in Fig.2, and the specific algorithmic flow is detailed in Alg.1 and Alg.2. In brief, at each communication round, the client acquires local knowledge through *local heterogeneous training and knowledge decoupling*, and decoupling it into a shared parameter subspace and a personalized parameter subspace (steps ①–③). Subsequently, the server performs *heterogeneous aggregation and compression* on the shared subspace parameters (steps

④–⑥). Finally, the shared and personalized subspaces are reconstructed to initiate a new round of personalized learning (steps ⑦–⑧). During the inference phase, we have developed an *dynamic weighted inference* mechanism that dynamically adjusts the knowledge ratio between the heterogeneous shared and personalized subspaces, thereby further improving the quality of responses to previously unseen tasks. This section will provide a detailed introduction.

$$
\Delta W = \begin{bmatrix} b_{11}, b_{12}, \underline{b_{13}} \\ b_{21}, b_{22}, \underline{b_{23}} \\ b_{31}, b_{32}, \underline{b_{33}} \end{bmatrix} \cdot \begin{bmatrix} a_{11}, a_{12}, a_{13} \\ a_{21}, a_{22}, a_{23} \\ \underline{a_{31}, a_{32}, a_{33}} \end{bmatrix}
$$

$$
= \begin{bmatrix} b_{11}a_{11} + b_{12}a_{21} + \underline{b_{13}a_{31}}, b_{11}a_{12} + b_{12}a_{22} + \\ b_{21}a_{11} + b_{22}a_{21} + \underline{b_{23}a_{31}}, b_{21}a_{12} + b_{22}a_{22} + \\ b_{31}a_{11} + b_{32}a_{21} + \underline{b_{33}a_{31}}, b_{31}a_{12} + b_{32}a_{22} + \end{bmatrix}
$$

$$
\begin{bmatrix} \underline{b_{13}a_{32}}, b_{11}a_{13} + b_{12}a_{23} + \underline{b_{13}a_{33}} \\ \underline{b_{23}a_{32}}, b_{21}a_{13} + b_{22}a_{23} + \underline{b_{23}a_{33}} \\ \underline{b_{33}a_{32}}, b_{31}a_{13} + b_{32}a_{23} + \underline{b_{33}a_{33}} \end{bmatrix}
$$

$$
= \begin{bmatrix} \overbrace{b_{11}, b_{12}}^{\text{shared}} \\ b_{21}, b_{22} \\ b_{31}, b_{32} \end{bmatrix} \begin{bmatrix} \overbrace{a_{11}, a_{12}, a_{13}}^{\text{shared}} \\ a_{21}, a_{22}, a_{23} \end{bmatrix}
$$

$$
+ \begin{bmatrix} \overbrace{b_{13}}^{\text{personalized}} \\ b_{23} \\ b_{33} \end{bmatrix} \begin{bmatrix} \overbrace{a_{31}, a_{32}, a_{33}}^{\text{personalized}} \end{bmatrix}
$$

$$
= \underbrace{B_s A_s}_{\text{shared knowledge}} + \underbrace{B_p A_p}_{\text{personalized calibration}} \tag{3}
$$

### 4.2.1. LOCAL HETEROGENEOUS TRAINING AND KNOWLEDGE DECOUPLING

During the initial round, the client $c \in \{1, 2, \ldots, c, \ldots, C\}$ configures the heterogeneous local LoRA settings, initializes the its parameters $B_c^0 \in \mathbb{R}^{d \times r_c}$ and $A_c^0 \in \mathbb{R}^{r_c \times k}$. Here, $r_c$ denotes the rank of client $c$'s LoRA matrix, while $d$ and $k$ represent the dimensions of the matrix, $r_c \ll d$, $r_c \ll k$, $C$ is the total number of all clients. In subsequent training rounds $t$, the client $c$ utilizing its local task data $D_c$, freeze the pre-trained model parameters and fine-tune only the LoRA module to learn local knowledge (step ①). This maps the local knowledge onto the parameter subspaces $B_c^t \in \mathbb{R}^{d \times r_c}$ and $A_c^t \in \mathbb{R}^{r_c \times k}$ of LoRA, as shown in Eq.4.

$$
A_c^t = A_c^{t-1} - \eta \nabla_A \mathcal{L}_c \big(A_c^{t-1}, B_c^{t-1}\big)
$$
$$
B_c^t = B_c^{t-1} - \eta \nabla_B \mathcal{L}_c \big(A_c^{t-1}, B_c^{t-1}\big), \tag{4}
$$

Subsequently, the client may customize personalized and generalized configurations according to its specific requirements. Assuming the configuration is $r_{c,p} \in (0, r_c)$, $r_{c,s} = r_c - r_{c,p}$. Based on it, the LoRA subspaces are decoupled into shared subspaces $B_{c,s}^t \in \mathbb{R}^{d \times r_{c,s}}$ and $A_{c,s}^t \in \mathbb{R}^{r_{c,s} \times k}$, and personalized subspaces $B_{c,p}^t \in \mathbb{R}^{d \times r_{c,p}}$ and

$A_{c,p}^t \in \mathbb{R}^{r_{c,p} \times k}$ (step ②). The relationship between $A_c^t$, $A_{c,s}^t$, $A_{c,p}^t$, $B_c^t$, $B_{c,s}^t$, and $B_{c,p}^t$ are as defined in Eq.5.

$$
A_c^t \to A_{c,s}^t \oplus A_{c,p}^t, \quad B_c^t \to B_{c,s}^t \oplus B_{c,p}^t
$$
$$
B_c^t A_c^t = B_{c,s}^t A_{c,s}^t + B_{c,p}^t A_{c,p}^t \quad \text{s.t.} r_c = r_{c,s} + r_{c,p}, \tag{5}
$$

where $r_{c,s}$ and $r_{c,p}$ denote the rank of the shared subspace and personalized subspace matrices of client $c$, respectively. $\oplus$ denotes the concatenation of two matrices. Notably, a higher $r_{c,s}$ value indicates more shared subspace parameters, reflecting a greater amount of shared knowledge.

The shared subspaces are then transmitted to the cloud server to participate in heterogeneous knowledge sharing and fusion with other clients (steps ③-⑤). The server will then send the fused shared knowledge $B_{c,s}^{\tilde{t}} \in \mathbb{R}^{d \times r_{c,s}}$ and $A_{c,s}^{\tilde{t}} \in \mathbb{R}^{r_{c,s} \times k}$ back to the clients (step ⑥). They will be integrated with the local personalized subspace knowledge, and reconstructed to $B_c^t$ and $A_c^t$ (step ⑦), as shown in Eq.6.

$$
A_c^t \leftarrow A_{c,s}^{\tilde{t}} \oplus A_{c,p}^t, B_c^t \leftarrow B_{c,s}^{\tilde{t}} \oplus B_{c,p}^t
$$
$$
B_c^t A_c^t = \Delta \tilde{W}_{c,s}^t + \Delta W_{c,p}^t = B_{c,s}^{\tilde{t}} A_{c,s}^{\tilde{t}} + B_{c,p}^t A_{c,p}^t. \tag{6}
$$

At this stage, the local LoRA matrix is reconstructed by integrating personalized subspace parameters into the shared subspace, enabling localized perturbations. The resulting LoRA subspace simultaneously captures shared knowledge from other clients and client-specific biases. The reconstructed parameters $B_c^t$ and $A_c^t$ are then used to initialize the next round of fine-tuning (step ⑧). By decoupling subspace parameters, personalized representations and generalized representations are confined to a low-rank subspace within the local LoRA matrix, enabling each client to adapt its local model through personalized and shared updates. Crucially, unlike existing approaches, it does not require retraining personalized components, thereby reducing computational overhead while avoiding overfitting to local tasks.

### 4.2.2. GENERALIZED KNOWLEDGE HETEROGENEOUS AGGREGATION AND COMPRESSION

After each client transmits its heterogeneous shared subspace parameters $A_{c,s}^t$ and $B_{c,s}^t$ to the cloud server, the server aggregates these parameters to fuse shared knowledge across all participating clients. First, we compute the low-rank shared updates as $\Delta W_{c,s}^t = B_{c,s}^t A_{c,s}^t$ ($W_{c,s}^t \in \mathbb{R}^{d \times k}$), which map the shared knowledge into the full rank parameter space. It is worth noting that, from the perspective of LoRA update, the effective model update is given by $B_{c,s}^t A_{c,s}^t$ rather than the individual matrices $B_{c,s}^t$ or $A_{c,s}^t$.

Subsequently, We aggregates all full rank shared subspaces from clients to construct a global shared subspace $\Delta W_{g,s}^t \in \mathbb{R}^{d \times k}$ (step ④). Since the shared subspaces $B_{c,s}^t$ and $A_{c,s}^t$ of each client may have heterogeneous ranks, the resulting information density in $\Delta W_{c,s}^t$ is also heterogeneous. Higher

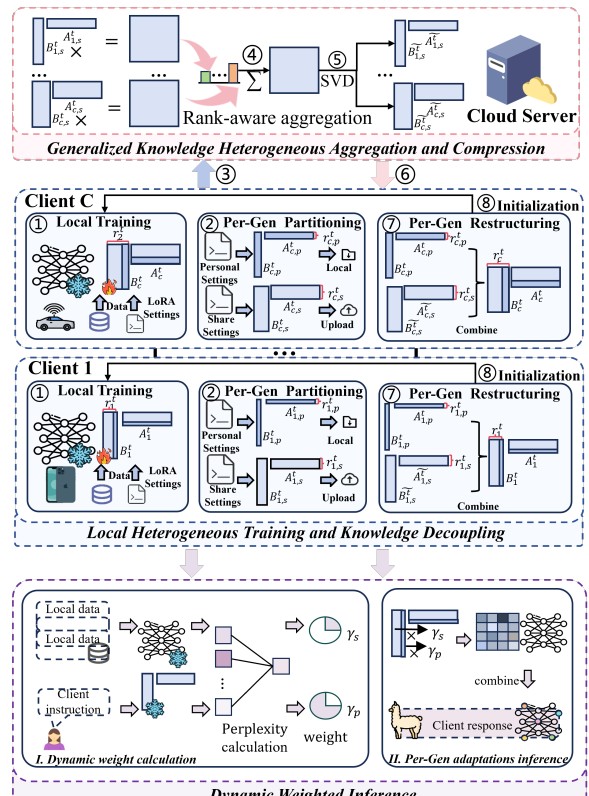

*Figure 2.* Het-CPFLoRA.

$r_{c,s}$ values enable the full rank shared subspace to encode richer shared knowledge, whereas lower $r_{c,s}$ values correspond to lower information density in the shared subspace. Determining aggregation weights solely based on training data volume overlooks this heterogeneity and may lead to unfair aggregation outcomes. To address this, we jointly balances the rank of the shared subspace and the scale of local data (Eq.7). As a result, clients with higher $r_{c,s}$ values are assigned larger aggregation weights, ensuring fairness.

$$\Delta W_{g,s}^t = \sum_{c=1}^{C} \alpha_c^t B_{c,s}^t A_{c,s}^t, \alpha_c^t = \frac{|D_c^t| \times r_{c,s}}{\sum_{c=1}^{C} |D_c^t| \times r_{c,s}}, \quad (7)$$

where $\alpha_c^t$ is the aggregation weight.

Finally, to accommodate the heterogeneity of shared subspaces across different clients, we apply singular value decomposition (SVD) to the global full rank shared subspace parameter $\Delta W_{g,s}^t$ (step ⑤). This decomposition compresses $\Delta W_{g,s}^t$ back to its corresponding low-rank representation, yielding the global low-rank shared subspace parameters $B_{c,s}^{\tilde{t}}$ and $A_{c,s}^{\tilde{t}}$ (Eq.8). The resulting $B_{c,s}^{\tilde{t}}$ and $A_{c,s}^{\tilde{t}}$ are then transmitted to the corresponding clients as global shared knowledge for subsequent local training.

$$U, \Omega, V^\top = SVD(\Delta W_{g,s}^t), A_{c,s}^{\tilde{t}} = V[: r_{c,s}, :]^\top$$
$$B_{c,s}^{\tilde{t}} = U[:, : r_{c,s}]\Omega[: r_{c,s}, : r_{c,s}]. \quad (8)$$

where $U, V, \Omega$ are the left singular vectors matrix, the right singular vectors matrix and the singular value matrix.

### 4.2.3. DYNAMIC WEIGHTED INFERENCE

The coupling between personalized and generalized knowledge in trained heterogeneous local models is typically fixed, which can limit response quality on locally unseen tasks. In practice, clients may provide inputs that deviate from the local task distribution, expecting high quality outputs. A fixed coupling pattern limits the adaptability under such shifts, causing degraded performance. We find that adjusting the proportions of shared and personalized subspaces mitigates this issue, better aligning the model with varying user instructions. As shown in Eq.9, when $\gamma_s$ is relatively high, the local model emphasizes shared knowledge, resulting in superior performance on locally unseen data. Conversely, it emphasizes personalized knowledge, demonstrating greater effectiveness in addressing the locally task.

$$W = W_0 + \gamma_s \Delta W_s + \gamma_p \Delta W_p \quad \text{s.t.} \gamma_s + \gamma_p = 1, \quad (9)$$

where $\gamma_s$ and $\gamma_p$ denote the coupling weight for the shared subspaces and the personalized subspaces.

To address this, we propose a dynamic weighted inference mechanism that adjusts the coupling between personalized and shared subspace parameters according to user text during inference. We formulate this as an OOD detection problem by assessing whether the user data distribution aligns with the local task distribution. When the distributions are similar, the personalized subspace is assigned a higher weight; otherwise, its influence is reduced.

Inspired by works as (Wu et al., 2023; Gangal et al., 2020), we use the perplexity of the text in the local model as the OOD detection metric. First, we compute the perplexity of the user data ($P_u$) and the average perplexity for the local dataset ($P_c$). Next, we calculate the average relative distance between the perplexities as the user data OOD score score$_c$ (Eq.10). A lower score indicates a higher likelihood that the data is out-of-distribution, and vice versa. To convert this score into coupling weights, we combine it with a scaling factor to form the personalized weights $\gamma_{c,p}$ (Eq.11). Notably, the scaling factor is computed from $r_{c,p}$ and $r_{c,s}$, aiming to adjust the coupling weights to account for the difference in parameter counts between the personalized and shared subspaces. Finally, we weight the parameters of both subspaces to obtain the local coupling model parameters $W_c$, which are used for inference to generate the response.

$$\text{score}_c = 1 - \frac{|P_u - P_c|}{\max(P_u, P_c)}, \quad (10)$$

$$\gamma_{c,p} = \frac{\text{score}_c \cdot \max(r_{c,p}, r_{c,s})}{(r_{c,p} + r_{c,p})}, \gamma_{c,s} = 1 - \gamma_{c,p}, \quad (11)$$

$$\begin{cases} A_{c,s} \leftarrow A_{c,s}^T, B_{c,s} \leftarrow \gamma_{c,s} B_{c,s}^T \\ A_{c,p} \leftarrow A_{c,p}^T, B_{c,p} \leftarrow \gamma_{c,p} B_{c,p}^T, \end{cases} \quad (12)$$

$$\Delta W_c = [B_{c,s}, B_{c,p}] \begin{bmatrix} A_{c,s} \\ A_{c,p} \end{bmatrix}, W_c = W_0 + \Delta W_c. \quad (13)$$

Through this, the local model dynamically adjusts the coupling weights of the personalized subspaces based on user data during inference. If the data closely resembles the local data distribution, the bias toward personalized knowledge in the local model is amplified. Conversely, the weights are reduced to limit the bias toward personalized knowledge.

### 4.3. Generalization Theoretical Analysis

We analyze the generalization ability of Het-CPFLoRA by referencing the discussion on generalization ability in (Bai et al., 2024), which extends Baxter's learning model. The expected loss is defined as $\mathcal{L}(\Delta W_g) \triangleq \mathbb{E}_{(x,y)\sim\mathcal{D}_c} f(\Delta W_g; (x,y))$. Note that $h_{\Delta W}()$ denotes the model-generated hypothesis of the LoRA weights $\Delta W$, while $f(\Delta W; (x,y))$ represents the loss function for a single data point $(x, y)$. Our analysis primarily relies on the following two key assumptions:

***Assumption 1.*** The following Lipschitz conditions hold: $|f(\Delta W; x, y) - f(\Delta W'; x, y)| \leq L_f\|\Delta W - \Delta W'\|$ and $\|h(\Delta W; x) - h(\Delta W'; x)\| \leq L_h\|\Delta W - \Delta W'\|$.

***Assumption 2.*** LoRA weights can be bounded in a ball with radius $R$. The error of the SVD approximation for each matrix is bounded, i.e., $\|SVD(\Delta W_g, r_c) - \Delta W_g\| \leq \sigma_c$.

Assumption 1 in works such as (Li et al., 2019; Ning et al., 2025), which are commonly used in FL, assumes that both $f$ and $h$ are Lipschitz continuous with respect to $\Delta W$, thereby ensuring the stability of the loss landscape. Assumption 2 in model generalization analyses, as discussed in works such as (Chen et al., 2023; Shamsian et al., 2021), is widely adopted to constrain the parameter space. The loss bound assumption for matrix low rank approximation is referenced in (Bai et al., 2024). Furthermore, the bound for SVD decompositions can also be derived from Eckart-Young-Mirsky theory (Eckart & Young, 1936). For the LoRA matrices, $\Delta W_{c,s}$ and $\Delta W_{c,p}$ are mutually independent. Only the SVD truncation of $\Delta W_{c,s}$ incurs approximation error, and $\Delta W_{c,p}$ introduces none. Therefore, based on these assumptions, we present the following theorem to demonstrate the generalization capability.

***Theorem 1.*** Let $SVD(\Delta W_g, r_c)$ denote the SVD function approximating $\Delta W_g$, and $r_c$ denote the decomposition rank. Under Assumptions 1 and 2, with probability at least $1 - \delta$, there exists a sample size $\hat{N} = \mathcal{O}(\frac{dk}{|C|\epsilon^2}\log(\frac{RL_fL_h}{\epsilon-2\sigma_cL_fL_h}) - \frac{\log\delta}{|C|\epsilon^2})$, such that for all $\Delta W_g$, the bound $\|\mathcal{L}(\Delta W_g) - \mathcal{L}(\Delta W_g')\| \leq \epsilon$ holds when the number of local data samples for each client $c$ exceeds $\hat{N}$.

Proof of Theorem 1 is provided in Appendix A. Our generalization bound depends on the LoRA rank $r_c$ and the number of clients ($|C|$). For the SVD approximation error bound $\sigma_c$, increasing the $r_c$ improves the approximation

accuracy, which reduces the error and consequently lowers the required sample size $\hat{N}$ for effective generalization.

## 5. Experiment

### 5.1. Experimental Setup

**Datasets and metric**. We use two datasets based on (Long et al., 2024), derived from the FLAN instruction dataset (Wei et al., 2021), which includes eight natural language processing tasks. For federated fine-tuning across heterogeneous tasks, each task is assigned to a corresponding client. During testing, we evaluate the local model's generalization and personalization capabilities using 1600 unseen samples from all eight tasks. Rouge-1 (Lin, 2004) is used as our performance metric. Details in Appendix C.1.2.

**Model and baselines**. We employ the Llama 3.2 1B model (Grattafiori et al., 2024) as the pre-trained model. To evaluate the excellence of our approach, we adopt the latest researches from recent years, FedDPA (Long et al., 2024), PF2LoRA (Hao et al., 2025), pFedGPT (Shen et al., 2025), and pFedSeq (Peng et al., 2025) as baselines. Details are provided in Appendix C.1.3.

**Experimental parameter settings**. The total number of federated learning rounds is 10, with eight clients participating. Following FedDPA (Long et al., 2024), we inject the LoRA matrix at the $q$ and $v$ vectors in each attention layer. Specifically, the LoRA rank is 8, and the scaling factor is 16, the batch size is 128, the learning rate is 3e-4.

### 5.2. Main Experiments

The experiment consists of three settings: *All homogeneous*, *Half heterogeneous*, and *All heterogeneous*, to demonstrate the effectiveness and superiority of Het-CPFLoRA. In the all homogeneous setting, both the local LoRA rank ($r_c = 8$) and the personalized subspace rank ($r_{c,p} = 2$) are the same across all clients. In the half heterogeneous setting, $r_c$ is consistent among clients ($r_c = 8$), while all $r_{c,p}$ are heterogeneous. For heterogeneous personalization, $r_{c,p}$ are randomly generated for each client using a normal distribution, with values constrained to be smaller than $r_c$. In the all heterogeneous setting, both $r_{c,p}$ and $r_c$ are heterogeneous. These ranks are independently generated for each client based on a normal distribution, with the personalized rank being smaller than the local rank. All experimental results are presented in Tables 1 and 2.

#### 5.2.1. ALL HOMOGENEOUS

Compared to FedDPA, we achieve up to 30.55% improvement and an average of 12.92% improvement in local task personalization on Dataset 1 without dynamic inference. Generalization capability improves by 6.04%. On Dataset

*Table 1.* Personalization capability (client-specific task) and generalization capability (all tasks) across all clients on Dataset 1.

| Dataset 1 | Test-Time Personalization | | | | | | | | | Test-Time Generalization |
|---|---|---|---|---|---|---|---|---|---|---|
| **Method** | **Para-phrase** | **Entail-ment** | **Structure to Text** | **Text For-matting** | **Linguistic Acc** | **Word Dis** | **Core-ference** | **Question CLS** | **Avg-Per** | **Avg-Gen** |
| *All Homogeneous* | | | | | | | | | | |
| PF2LoRA | 43.42 | 29.95 | 48.76 | 81.21 | 36.55 | 51.77 | 51.23 | 50.87 | 49.22 | 48.48 |
| pFedGPT | 55.32 | 31.19 | 53.39 | 83.70 | 41.04 | 52.49 | 56.22 | 59.47 | 54.13 | 52.18 |
| pFedSeq | 56.96 | 33.09 | 55.63 | 92.88 | 42.16 | 52.54 | 60.37 | 61.40 | 56.88 | 52.41 |
| FedDPA | 24.08 | 30.17 | 44.85 | 72.89 | 30.97 | 36.98 | 46.13 | 32.90 | 39.87 | 32.44 |
| FedDPA-Auto | 55.55 | 29.25 | 53.36 | **94.22** | **42.50** | 52.00 | 59.65 | 64.00 | 56.32 | 54.37 |
| Our | 54.63$_{\uparrow 30.55}$ | 31.61$_{\uparrow 1.44}$ | 53.86$_{\uparrow 9.01}$ | 79.36$_{\uparrow 6.47}$ | 38.75$_{\uparrow 7.78}$ | 50.79$_{\uparrow 13.81}$ | 58.46$_{\uparrow 12.33}$ | 54.93$_{\uparrow 22.03}$ | 52.79$_{\uparrow 12.92}$ | 38.48$_{\uparrow 6.04}$ |
| Our-auto | **60.00**$_{\uparrow 3.04}$ | **33.50**$_{\uparrow 0.41}$ | **58.64**$_{\uparrow 3.01}$ | 93.27 | 39.15 | **53.00**$_{\uparrow 0.46}$ | **61.91**$_{\uparrow 1.54}$ | **73.50**$_{\uparrow 9.50}$ | **59.12**$_{\uparrow 2.24}$ | **54.72**$_{\uparrow 0.35}$ |
| *Half Heterogeneous* | | | | | | | | | | |
| Our | 56.85$_{\uparrow 32.77}$ | 32.53$_{\uparrow 2.36}$ | 54.59$_{\uparrow 9.74}$ | 83.80$_{\uparrow 10.91}$ | 43.25$_{\uparrow 12.28}$ | 52.02$_{\uparrow 15.04}$ | 53.29$_{\uparrow 7.16}$ | 57.61$_{\uparrow 24.71}$ | 54.24$_{\uparrow 14.37}$ | 38.67$_{\uparrow 6.23}$ |
| Our-auto | **61.50**$_{\uparrow 4.54}$ | **33.57**$_{\uparrow 0.48}$ | **59.35**$_{\uparrow 3.72}$ | 94.16 | 44.55$_{\uparrow 2.05}$ | 52.98$_{\uparrow 0.44}$ | **62.01**$_{\uparrow 1.64}$ | 67.50 | **59.45**$_{\uparrow 2.57}$ | **55.13**$_{\uparrow 0.76}$ |
| *All Heterogeneous* | | | | | | | | | | |
| Our | 55.66$_{\uparrow 31.58}$ | 29.97 | 55.83$_{\uparrow 10.98}$ | 88.95$_{\uparrow 16.06}$ | 46.50$_{\uparrow 15.53}$ | 49.52$_{\uparrow 12.54}$ | 59.78$_{\uparrow 13.65}$ | 68.32$_{\uparrow 35.42}$ | 56.82$_{\uparrow 16.95}$ | 43.43$_{\uparrow 10.99}$ |
| Our-auto | **61.32**$_{\uparrow 4.36}$ | **34.42**$_{\uparrow 1.34}$ | **61.33**$_{\uparrow 5.70}$ | **94.97**$_{\uparrow 0.75}$ | **49.50**$_{\uparrow 7.00}$ | **54.00**$_{\uparrow 1.46}$ | **64.69**$_{\uparrow 4.32}$ | **74.50**$_{\uparrow 10.50}$ | **61.84**$_{\uparrow 4.96}$ | **56.17**$_{\uparrow 1.80}$ |

*Table 2.* Personalization capability (client-specific task) and generalization capability (all tasks) across all clients on Dataset 2.

| Dataset 2 | Test-Time Personalization | | | | | | | | | Test-Time Generalization |
|---|---|---|---|---|---|---|---|---|---|---|
| **Method** | **Para-phrase** | **Common-sense** | **Entail-ment** | **Text For-matting** | **Summari-zation** | **Reading Com** | **Senti-ment** | **Open QA** | **Avg-Per** | **Avg-Gen** |
| *All Homogeneous* | | | | | | | | | | |
| PF2LoRA | 46.44 | 34.09 | 39.95 | 40.33 | 21.08 | 32.14 | 69.49 | 55.35 | 43.73 | 42.42 |
| pFedGPT | 49.02 | 37.81 | 42.19 | 43.15 | 21.94 | 33.26 | 70.55 | 59.30 | 44.66 | 43.06 |
| pFedSeq | 49.41 | 37.86 | 43.06 | 45.37 | 22.09 | 34.90 | 71.92 | 61.04 | 45.71 | 44.63 |
| FedDPA | 13.30 | 30.21 | 22.30 | 32.13 | 16.68 | 14.51 | 59.91 | 41.94 | 28.87 | 23.47 |
| FedDPA-Auto | 49.07 | 37.97 | 42.18 | 43.90 | 22.26 | 34.79 | 76.50 | 60.26 | 45.86 | 45.32 |
| Our | 32.91$_{\uparrow 19.61}$ | 37.95$_{\uparrow 7.74}$ | 44.06$_{\uparrow 21.76}$ | 51.87$_{\uparrow 19.74}$ | 23.70$_{\uparrow 7.02}$ | 29.57$_{\uparrow 15.06}$ | 65.71$_{\uparrow 5.80}$ | 51.24$_{\uparrow 9.30}$ | 42.12$_{\uparrow 13.25}$ | 31.97$_{\uparrow 8.50}$ |
| Our-auto | **53.00**$_{\uparrow 3.59}$ | **39.69**$_{\uparrow 1.72}$ | **56.63**$_{\uparrow 13.57}$ | **58.76**$_{\uparrow 13.39}$ | **26.38**$_{\uparrow 4.12}$ | **45.29**$_{\uparrow 10.39}$ | 69.50 | **68.92**$_{\uparrow 7.88}$ | **52.27**$_{\uparrow 6.41}$ | **45.84**$_{\uparrow 0.52}$ |
| *Half Heterogeneous* | | | | | | | | | | |
| Our | 38.70$_{\uparrow 25.40}$ | 37.25$_{\uparrow 7.04}$ | 51.33$_{\uparrow 29.03}$ | 64.42$_{\uparrow 32.29}$ | 26.42$_{\uparrow 9.74}$ | 31.76$_{\uparrow 17.25}$ | 63.01$_{\uparrow 3.10}$ | 53.33$_{\uparrow 11.39}$ | 45.77$_{\uparrow 16.90}$ | 32.33$_{\uparrow 8.86}$ |
| Our-auto | **52.50**$_{\uparrow 3.09}$ | **39.30**$_{\uparrow 1.33}$ | **51.72**$_{\uparrow 8.66}$ | **67.60**$_{\uparrow 22.23}$ | **29.43**$_{\uparrow 7.17}$ | **55.71**$_{\uparrow 20.81}$ | 74.50 | **66.91**$_{\uparrow 5.87}$ | **54.71**$_{\uparrow 8.85}$ | **46.13**$_{\uparrow 0.81}$ |
| *All Heterogeneous* | | | | | | | | | | |
| Our | 37.97$_{\uparrow 24.67}$ | 39.16$_{\uparrow 8.95}$ | 46.02$_{\uparrow 23.72}$ | 63.69$_{\uparrow 31.56}$ | 27.56$_{\uparrow 10.88}$ | 35.10$_{\uparrow 20.59}$ | 64.64$_{\uparrow 4.73}$ | 59.57$_{\uparrow 17.63}$ | 46.71$_{\uparrow 17.84}$ | 37.19$_{\uparrow 13.72}$ |
| Our-auto | **57.01**$_{\uparrow 7.60}$ | **42.47**$_{\uparrow 4.50}$ | **54.49**$_{\uparrow 11.43}$ | **80.64**$_{\uparrow 35.27}$ | **30.41**$_{\uparrow 8.15}$ | **63.54**$_{\uparrow 28.64}$ | 73.00 | **70.00**$_{\uparrow 8.96}$ | **58.95**$_{\uparrow 13.09}$ | **47.23**$_{\uparrow 1.91}$ |

2, personalization improves by 13.25%, and generalization by 8.50%. With dynamic weighted inference, personalization performance on Dataset 1 increases by up to 9.50% compared to all baselines, with an average gain of 2.24% and a 0.35% improvement in generalization. On Dataset 2, personalization improves by 6.41%, with a 0.52% boost in generalization. Overall, our approach outperforms all baselines in both personalization and generalization across most heterogeneous tasks, with performance gains of about 10% in question answering and text formatting tasks. These results show that single adapter training mitigates optimization conflicts in dual adapter iterative training, while parameter decoupling tightens the integration of personalized and generalized knowledge, reducing knowledge loss and significantly enhancing local model capabilities.

### 5.2.2. HALF HETEROGENEOUS

We only present our results, as existing methods do not support heterogeneous settings. Compared to homogeneous settings, the average personalized capability improved by 1.45% (Dataset 1) and 3.65% (Dataset 2) under non-dynamic weighted inference. Under dynamic weighted inference, the average personalized capability improved by 0.33% (Dataset 1) and 2.44% (Dataset 2). Heterogeneous personalization settings allow clients to independently adjust the proportion of shared and personalized knowledge, enabling them to capture richer personalized representations than homogeneous settings. Meanwhile, the average generalization performance on both datasets remains comparable to that of the homogeneous setting.

### 5.2.3. ALL HETEROGENEOUS

Without dynamic weighted inference, Het-CPFLoRA outperforms FedDPA, with average personalization improvements of 16.95% on Dataset 1 and 17.84% on Dataset 2, and generalization gains of 10.99% and 13.70%, respectively. Compared to the homogeneous setting, personalization improves by 4.03% and 4.59%, while generalization increases by 4.95% and 5.22%. Compared to the half heterogeneous setting, personalization improves by 2.58%

and 0.94%, and generalization by 4.76% and 4.86%. With dynamic weighted inference, Het-CPFLoRA consistently outperforms all baselines, showing average personalization gains of 4.96% on Dataset 1 and 13.09% on Dataset 2, and generalization gains of 1.80% and 1.91%. Compared to the homogeneous setting, personalization improves by 2.72% and 6.68%, and generalization increases by 1.45% and 1.39%.Compared to the half heterogeneous setting, personalization improves by 2.19% and 4.24%, with generalization gains of 1.04% and 1.10%.

Fully heterogeneous configurations address both heterogeneous client resources and personalization requirements. Here, Het-CPFLoRA outperforms both half-heterogeneous and homogeneous settings in personalization and generalization, offering greater personalized parameter flexibility and richer knowledge sharing, resulting in superior performance in highly heterogeneous scenarios.

### 5.2.4. THE IMPACT OF PERSONALIZED RANK

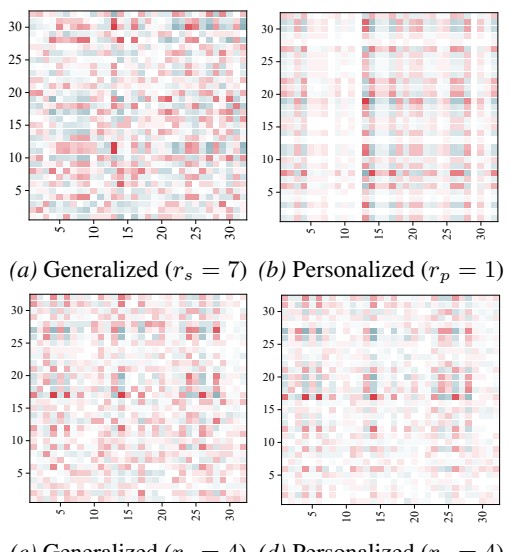

*(a) Generalized ($r_s = 7$)  (b) Personalized ($r_p = 1$)*

*(c) Generalized ($r_s = 4$)  (d) Personalized ($r_p = 4$)*

*Figure 3.* Heatmaps of parameters for full rank generalized subspaces and personalized subspaces under different ranks.

We analyze the impact of personalization rank on the local model's personalization and generalization performance with a fixed local LoRA rank of 8 and client personalization ranks ($r_p$) set to 1, 2, 3, and 4. The results are shown in Table 3. We sum each client's specific task scores and then average them as the average personalized score (*Avg-Per*). Similarly, we sum each client's performance over all tasks and average them as the average generalized score (*Avg-Gen*). Table 3 indicates that increasing the proportion of parameters in the personalized subspace induces a clear trade-off. *Avg-Per* is lowest at a 7 : 1 split and highest at a 4 : 4 split, with an overall increasing tendency. Conversely, *Avg-Gen* is highest at 7 : 1 and lowest at 4 : 4, with an

overall decreasing tendency. This confirms that $r_p$ is a key factor in regulating the personalization and generalization abilities of local models.

To illustrate how personalization rank affects LoRA parameters, we visualize heatmaps of the full rank personalization parameters and local model parameters across different personalization ranks, as shown in Fig.3. We focus on the full rank product $BA$, which directly corresponds to the effective weight update applied to the local model. It shows that when the personalized subspace is small, the full rank personalization parameters are sparse, limiting their impact on the overall LoRA update. Consequently, the local model relies more on the shared component and emphasizes generalized knowledge. In contrast, with a higher proportion of personalized parameters, the full rank personalization parameters become denser, exerting a stronger influence on the LoRA update and increasing the focus on client specific personalized knowledge. This highlights that decoupling personalization and generalization through varying $r_p$ significantly affects the capabilities of local models.

*Table 3.* The impact of different personalized subspace ranks.

| Personalized subspace rank | Dataset 1 | | Dataset 2 | |
| vs Shared subspace rank | Avg-Per | Avg-Gen | Avg-Per | Avg-Gen |
|---|---|---|---|---|
| 1:7 | 54.22 | 43.13 | 42.21 | 33.00 |
| 2:6 | 54.53 | 40.64 | 45.40 | 31.75 |
| 3:5 | 56.87 | 39.06 | 45.66 | 30.97 |
| 4:4 | 57.64 | 38.63 | 46.18 | 30.43 |
| - | ↗ | ↘ | ↗ | ↘ |

*Table 4.* The impact of different decoupling strategies.

| Decoupling strategies | Avg-Per | Avg-Gen |
|---|---|---|
| Front position | 52.27 | 45.84 |
| Rear position | 52.88 | 45.60 |
| Semi-random position | 52.09 | 45.71 |

## 5.3. Ablation Study

### 5.3.1. DECOUPLING STRATEGY

The decoupling strategy employed in the main experiment involves selecting the last row or column of the $B$ and $A$ matrices as the personalized subspace (Rear position strategy). To assess the impact of decoupling strategies, we conduct ablation experiments using front-position decoupling and semi-random decoupling. The front-position decoupling refer to selecting the first $r_p$ columns or rows of the $B$ and $A$ matrices as the personalization subspaces. Semi-random decoupling refers to randomly selecting columns $i$ through $j$ from matrix $B$ as the personalization space, whilst simultaneously selecting rows $i$ through $j$ from matrix $A$ as the personalization space (where $j - i = r_p - 1$). Note that fully random decoupling is not valid, as it involves randomly selecting rows and columns from matrices $B, A$ for decoupling (which violates the rules of matrix multiplication).

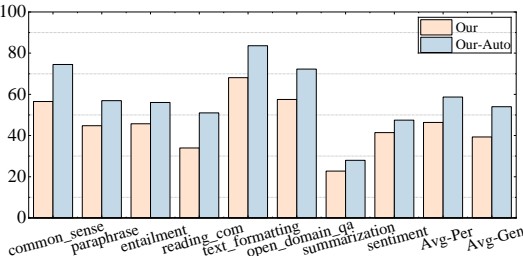

*Figure 4.* Performance on the Qwen model.

Taking $r_p = 4$, $r = 4$ and Dataset 2 as examples, the results in a homogeneous environment are shown in Table 4. The results indicate that there is little variation across different decoupling strategies. Valid decoupling strategies do not affect the effectiveness of algorithm training.

### 5.3.2. SCALING FACTOR

We conduct ablation experiments to compare different scaling factor designs, including no scaling, the scaling factor adopted by FedDPA, and our design. The scaling factor prevents excessively large weights, which may otherwise compromise both personalization and generalization. Unlike FedDPA's empirical design, our design explicitly incorporates the proportion of the personalization rank. Taking Dataset 2 and the heterogeneous setting as examples, the results obtained after conducting five rounds of testing and calculating the mean are shown in Table 5. These show that our design significantly improves generalization compared to other designs, confirming that the scaling factor mitigates over personalization and enhances generalization.

*Table 5.* The impact of scaling factor.

| Scaling factor | Avg-Per | Avg-Gen |
|---|---|---|
| None | 57.01 | 44.38 |
| FedDPA | 57.22 | 46.15 |
| Our | 58.91 | 47.49 |

### 5.3.3. OTHER MODEL

We conduct model experiments on Qwen2.5-1.5B and Dataset 2 to verify robustness across different model architectures. In heterogeneous environments, $Avg - Per$ and $Avg - Gen$ improve by 6.64% and 4.05%, respectively, compared with the baselines. Moreover, Fig.4 shows the score comparisons for each local task before and after introducing the dynamic weighting mechanism. Overall, These indicate that both the fine-tuning algorithm and the weighted inference mechanism remain robust on the Qwen model.

## 6. Additional Experiments

To comprehensively demonstrate scalability and effectiveness, we conduct the additional experiments.

**Scalability.** We conduct scalability experiments with larger

models and more clients, detailed in Appendix C.2. **Dynamic Weighted Inference.** We conduct an ablation analysis of the dynamic weighted mechanism during the inference stage, detailed in Appendix C.3. **Convergence Analysis.** We demonstrate and analyze the convergence of our algorithm, detailed in Appendix C.4. **Overhead**. We calculate the local computational overhead and communication overhead, detailed in Appendix C.5.

All results show that our approach is overhead lightweight, with stable convergence. It maintains model and clients scalability, consistently outperforming baselines.

## 7. Limitations and Future Work

Our research enables customizable heterogeneous personalized learning, but cannot determine the optimal subspace rank partition. We argue that local task complexity is a key in configuration optimization. The low complexity tasks require a smaller $r_p$, whereas high complexity tasks need a larger rank to capture personalized knowledge.

Compared with random settings, $Avg - Per$ and $Avg - Gen$ improve by 0.85% and 0.68%, respectively. With dynamic reasoning, the gains become 1.03% and 0.36%. These suggest that task complexity helps determine the optimal personalization rank partition.

However, task complexity alone is insufficient, as local computational resources, data quality may also affect local model performance. Future work will explore multi-factor rank partitioning to enable dynamic and adaptive personalized training under resource-constrained settings.

## 8. Conclusion

Personalized federated LoRA has emerged as a mainstream paradigm for fine-tuning LLMs. However, enabling client customization of heterogeneous personalization levels remains a key challenge for the scalability and practicality of PFL. If adopting multi-adapter will exacerbate optimization conflicts, limiting their effectiveness. Thus, this paper presents the first work on heterogeneous personalized FL that allows customization of both personalization and generalization levels, based on a single adapter. By leveraging the LoRA parameter decoupling property, our approach supports flexible adjustment of personalized and generalized knowledge during both training and inference, substantially improving practicality. Extensive evaluations across diverse NLP tasks demonstrate that our work consistently enhances both personalization and generalization performance, while exhibiting strong scalability in complex heterogeneous environments. Future work will investigate adaptive strategies for balancing personalized and generalized knowledge based on local data under resource constraints.

## Impact Statement

This work presents a federated learning algorithm designed to enable heterogeneous, customizable personalization on a foundational large language model. The algorithm is suitable for complex environments characterized by heterogeneous local resources, heterogeneous local data, and heterogeneous local personalization requirements. We do not anticipate this research having any other significant social implications.

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

# Appendix

## A. Generalization Theoretical Proof

In this section, we conduct a theoretical analysis of Theorem 1.

*Proof:* Let $\mathbb{H}^n$ denote the function space whose elements consist of LoRA weights $\Delta W \in \mathbb{R}^{d \times k}$. The distance metric $\lambda$ is defined as:

$$
\begin{aligned}
&\lambda(\Delta W_{g,c} - \Delta W'_{g,c}) \\
&= \lambda(\Delta W_{c,s} + \Delta W_{c,p} - \Delta W_{c,p} - \Delta W'_{c,s}) \\
&= \lambda(\Delta W_{c,s} - \Delta W'_{c,s}) \\
&= \frac{1}{n} \mathbb{E}_{x,y \sim D_c} \left[ \left| \sum \left( f(\Delta W_{c,s}; x, y) - \sum f(\Delta W'_{c,s}; x, y) \right) \right| \right] \\
&\overset{a_1}{\leq} L_f \| h_{\Delta W_{c,s}} - h_{\Delta W'_{c,s}} \| \\
&= L_f \| h(\text{SVD}(\Delta W_g, r^c)) - h(\text{SVD}(\Delta W'_g, r^c)) \| \\
&\overset{a_1}{\leq} L_f L_h \| \text{SVD}(\Delta W_g, r^c)) - \text{SVD}(\Delta W_g', r^c)) \| \\
&= L_f L_h \| \text{SVD}(\Delta W_g, r^c) - \Delta W_g - \text{SVD}(\Delta W_g', r^c) + \Delta W_g' \\
&\quad + \Delta W_g - \Delta W_g' \| \\
&\overset{a_2}{\leq} L_f L_h \| \text{SVD}(\Delta W_g, r^c) - \Delta W_g \| \\
&\quad + L_f L_h \| \text{SVD}(\Delta W_g', r^c) - \Delta W_g' \| \\
&\quad + L_f L_h \| \Delta W_g - \Delta W_g' \| \\
&\overset{a_2}{\leq} L_f L_h [2\sigma_c + \| \Delta W_g - \Delta W_g' \|]
\end{aligned}
\tag{14}
$$

where $a_1$ is *Assumption 1* and $a_2$ is *Assumption 2*. From the above proof, we can get an $\epsilon$-covering in metric $\lambda(\Delta W_{g,c} - \Delta W_{g,c}')$ if we select a covering in the parameter space with $\| \Delta W_g - \Delta W_g' \|$ equal to $\frac{\epsilon}{L_f L_h} - 2\sqrt{\sigma_c}$. Then, the covering number of $\mathbb{H}^{|C|}$, denoted as $\log \mathcal{B}(\epsilon, \mathbb{H}^{|C|}) = \mathcal{O}(dk \log(\frac{R L_f L_h}{\epsilon - 2\sigma_c L_f L_h}))$. Here, $R$ is the radius within which the LoRA weights are constrained in *Assumption 2*. $|C|$ is the total number of clients. $d$ and $k$ are the dimension of the LoRA weight $\Delta W_g$ parameter space.

According to research ([Wang et al., 2024b](); [Baxter, 2000](); [Chen et al., 2023]()), there exists $\hat{N}$ such that $\lambda(\Delta W_{g,c} - \Delta W_{g,c}') \leq \epsilon$ holds, and $\hat{N}$ is

$$
\begin{aligned}
\hat{N} &= \mathcal{O}\left( \frac{1}{|C|\epsilon^2} \log \frac{\mathcal{B}(\epsilon, \mathbb{H}^{|C|})}{\delta} \right) \\
&= \mathcal{O}\left( \frac{dk}{|C|\epsilon^2} \log\left( \frac{R L_f L_h}{\epsilon - 2\sigma_c L_f L_h} \right) - \frac{\log \delta}{|C|\epsilon^2} \right)
\end{aligned}
\tag{15}
$$

The above completes the proof of *Theorem 1*.

## B. Algorithm Description

The pseudocode for the algorithmic processes described in Section 4.2 is shown in Alg.1 and Alg.2. Alg.1 represents the Het-CPFLoRA algorithmic process, while Alg.2 represents the dynamic weighted inference process.

## C. Additional Experimental Details

### C.1. Experimental setup

#### C.1.1. EXPERIMENTAL ENVIRONMENT

All experiments are conducted using Python 3.10 and PyTorch on three NVIDIA GeForce RTX 4090 GPUs.

---

**Algorithm 1** Het-CPFLoRA.

---

**Input:** Local data $\{D_c\}_{c \in \{1,2,...,C\}}$, FL training rounds $T$, pre-training large model parameters $W_0$, local LoRA rank $\{r_c\}_{c \in \{1,2,...,C\}}$, shared parameter subspace rank $\{r_{c,s}\}_{c \in \{1,2,...,C\}}$, personalized parameter subspace rank $\{r_{c,p}\}_{c \in \{1,2,...,C\}}$, LoRA layer set $\bigcup_{l \in \{1,2,...,\mathcal{L}\}}$.
**Output:** LoRA parameters $\bigcup_{c \in \{1,2,...,C\}} \{B_c^T, A_c^T\}$ for each client.
**Initialization:**
Clients configure LoRA settings $\{r_c\}_{c \in \{1,2,...,C\}}$ and initialize parameters.
**for** $t = 1$ **to** $T$ **do**
  $Client$ :

      **for** $c$ **in** $\{1, 2, ..., C\}$ **do**
        Set the shared subspace and personalized subspace ranks $r_{c,s}$ and $r_{c,p}$;
        Download the shared LoRA subspace parameters $B_{c,s}^{\tilde{t}}$, $A_{c,s}^{\tilde{t}}$ from cloud server;          ◁ *Step ⑥*
        Combine personalized subspace parameters $B_{c,p}^t$, $A_{c,p}^t$ and shared subspace parameters $B_{c,s}^{\tilde{t}}$, $A_{c,s}^{\tilde{t}}$, initialize
        local LoRA: $B_c^t \leftarrow B_{c,s}^{\tilde{t}} \oplus B_{c,p}^t$, $A_c^t \leftarrow A_{c,s}^{\tilde{t}} \oplus A_{c,p}^t$;          ◁ *Steps ⑦ ⑧*
        $B_c^{t+1}$, $A_c^{t+1}$=LOCALTUNING($D_c$, $W_0$, $B_c^t A_c^t$);          ◁ *Step ①*
        Decoupling parameters $B_c^{t+1}$ and $A_c^{t+1}$ into the shared subspace $\{B_{c,s}^{t+1}, A_{c,s}^{t+1}\}$ and the personalized subspace
        $\{B_{c,p}^{t+1}, A_{c,p}^{t+1}\}$ based on $r_{c,s}$ and $r_{c,p}$;          ◁ *Step ②*
        Upload $\{B_{c,s}^{t+1}, A_{c,s}^{t+1}\}$, $r_{c,s}$ to server;          ◁ *Step ③*
      **end for**

  $Server$ :

      Server aggregation shared subspace parameters: $\Delta W_{g,s}^{t+1} \leftarrow Eq.7$;          ◁ *Step ④*
      **for** $c$ **in** $\{1, 2, ..., C\}$ **do**
        **for** $l$ **in** $\mathcal{L}$ **do**
          $\{B_{c,s,l}^{\tilde{t+1}}, A_{c,s,l}^{\tilde{t+1}}\} = SVD(\Delta W_{g,s,l}^{t+1}, r_{c,s})$;          ◁ *Step ⑤*
        **end for**
        $\{B_{c,s}^{\tilde{t+1}}, A_{c,s}^{\tilde{t+1}}\} = \bigcup_{l \in \mathcal{L}} \{B_{c,s,l}^{t+1}, A_{c,s,l}^{t+1}\}$.
      **end for**

**end for**
**return** $\bigcup_{c \in \{1,2,...,C\}} \{B_c^T, A_c^T\}$

---

---

**Algorithm 2** Dynamic Weighted Inference.

---

**Input:** Local data $D_c$, user input instruction, pre-training large model parameters $W_0$, local LoRA parameters $A_c = A_{c,s} \oplus A_{c,p}$, $B_c = B_{c,s} \oplus B_{c,p}$.
**Output:** Dynamically weighted LLM parameters $\Delta W_c$.
*Dynamic weights calculation:*

    Calculate the perplexity of the user input and the local data $P_u$, $P_c$;
    Calculate the OOD score for user data $score_c \leftarrow$ Eq.10;
    Calculating personalized and shared subspace coupling weights $\gamma_{c,p}, \gamma_{c,s} \leftarrow Eq.11$;

*Personalization and generalization adaptation:*

    Personalization and shared subspace parameter weighted $A_{c,s}^t, B_{c,s}^t, A_{c,p}^t, B_{c,p}^t \leftarrow Eq.12$;

    Calculation the local LLM parameter $W_c = W_0 + \Delta W_c = W_0 + \begin{bmatrix} B_{c,s}, B_{c,p} \end{bmatrix} \begin{bmatrix} A_{c,s} \\ A_{c,p} \end{bmatrix}$.

---

### C.1.2. DATASETS

Following (Long et al., 2024), each of our datasets contains 2400 data points, with 300 samples per task. Specifically, dataset 1 contains word disambiguation, text formatting, structure to text, question classification, paraphrase, linguistic acceptability, entailment, coreference tasks, while dataset 2 contains common sense, entailment, open domain question answering, paraphrase, reading comprehension, sentiment, summarization, text formatting tasks. Both test sets comprise 1600 training-unseen data points, covering all tasks from the corresponding training set.

### C.1.3. BASELINES

Our baseline approaches are as follows:

① FedDPA (Long et al., 2024). This method achieves personalized learning by locally iteratively training global and personalized adapters while sharing the global adapter.

② PF2LoRA (Hao et al., 2025). This method adaptively determines the predefined LoRA rank based on local data and completes learning through a two-stage iterative fine-tuning of global and personalized adapters.

③ pFedGPT (Shen et al., 2025). This method adjusts the weights between the global and personalized adapters via hierarchical Bayesian optimization search.

④ pFedSeq (Peng et al., 2025). This method utilizes a hypernetwork to process historical update information from the local adapter, generating personalized calibrations to fine-tune the global adapter.

### C.2. Scalability Experiments

To evaluate the effectiveness of the Het-CPFLoRA algorithm with larger models and a massive number of clients, we conduct experiments on both model scalability and client scalability.

### C.2.1. MODEL SCALABILITY

We scale the base model from Llama 1B to 3B and 7B parameters and performed federated fine-tuning, with the results shown in Fig.5. Across both datasets, the average performance and generalization capabilities of all client models showed an upward trend as the model parameters increased. This suggests that a larger number of parameters allows the model to more comprehensively capture knowledge across diverse heterogeneous tasks. Compared to the all baselines, our approach shows significant improvements in both personalization and generalization capabilities with larger models, highlighting its effective potential for extending foundational models.

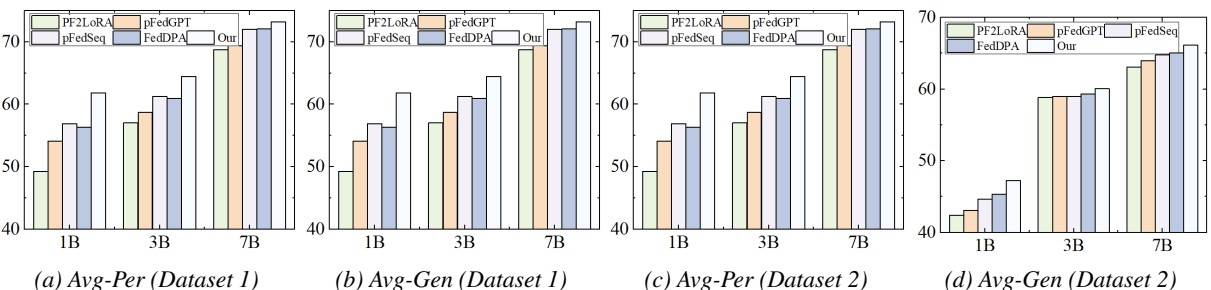

*(a) Avg-Per (Dataset 1)*  *(b) Avg-Gen (Dataset 1)*  *(c) Avg-Per (Dataset 2)*  *(d) Avg-Gen (Dataset 2)*

*Figure 5.* Comparison of average generalization and personalization capabilities in Llama models 1B, 3B, and 7B.

### C.2.2. CLIENT SCALABILITY

We randomly split each local dataset into six parts to scale the number of clients from 8 to 48, verifying the algorithm's scalability with a large number of clients. The results (see Table 6) show that Het-CPFLoRA retains its inherent superiority even with a large number of clients. Notably, compared to the experimental results in Tables 1 and 2, all methods exhibit a decline in both personalized and generalized performance. This decline occurs because, while the total dataset size remains unchanged, the significant reduction in local data per client limits the amount of knowledge the model can learn.

| Methods | Dataset 1 | | Dataset 2 | |
|---|---|---|---|---|
| | Avg-Per | Avg-Gen | Avg-Per | Avg-Gen |
| PF2LoRA | 52.35 | 50.16 | 44.18 | 44.10 |
| pFedGPT | 51.92 | 50.88 | 44.02 | 43.83 |
| pFedSeq | 53.77 | 52.11 | 45.81 | 45.02 |
| FedDPA | 52.13 | 51.01 | 44.75 | 44.59 |
| Our | **59.36** | **54.29** | **57.11** | **45.54** |
| Improve | 5.59 | 2.18 | 11.30 | 0.52 |

Table 6. Client scalability experiment.

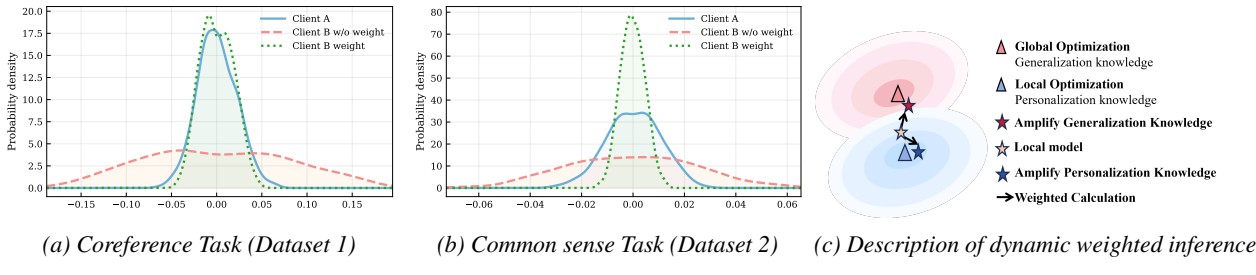

(a) Coreference Task (Dataset 1)   (b) Common sense Task (Dataset 2)   (c) Description of dynamic weighted inference

Figure 6. LoRA activation comparison.

## C.3. The Impact of Dynamic Weighted Inference

We conduct an ablation analysis of the dynamic weighting mechanism during the inference stage. Specifically, we compare our approach against two scenarios: one without weighting and the other using the cosine similarity method with encoded embeddings, as employed by FedDPA, as shown in Table 7. These experiments are conducted under homogeneous scenarios, as FedDPA only supports such settings. After applying the dynamic weighting, the local model's average personalization and generalization performance improve to 59.12% and 54.72%, and 52.27% and 45.87%, respectively, surpassing the baseline method. Additionally, compared to the non-dynamically weighted approach, personalization capability increases by 6%–10%, and generalization capability improves by 14%–16%.

| FedDPA | Our | Dataset 1 | | Dataset 2 | |
|---|---|---|---|---|---|
| | | Avg-Per | Avg-Gen | Avg-Per | Avg-Gen |
| ✗ | ✗ | 52.79 | 38.48 | 42.12 | 31.97 |
| ✓ | ✗ | 56.77 | **54.73** | 45.62 | 45.29 |
| ✗ | ✓ | **59.12** | 54.72 | **52.27** | **45.84** |

Table 7. The impact of dynamic weighted inference.

Additionally, we collect the last layer LoRA activations from the local model to more clearly illustrate the role of the dynamic weighting mechanism during inference. Specifically, take the coreference task in Dataset 1 and the open domain QA task in Dataset 2 as examples. During federated fine-tuning on Dataset 1, client A's local task is coreference, while client B's task is question classification. We collect activation data from the last LoRA layer of client A's local model (denoted as model A1) when inferring the sample 1 (coreference task). Simultaneously, we collect activation data from client B's local model (referred to as model B1) on the sample 1 with and without the dynamic weighting mechanism, as shown in Fig.6a. Similarly, for Dataset 2, client A's local task is open-domain question answering, while client B's task is common sense. We collect activation data from the final LoRA layer of client A's local model (referred to as model A2) on the sample 10 (open domain QA task). Simultaneously, we collect activation data from client B's local model (referred to as model B2) on the sample 10 with and without the dynamic weighting mechanism, as shown in Fig.6b. It is worth noting that we employ kernel density estimation (KDE) to process the activation data, thereby obtaining the probability density distribution of the activations, as shown in Fig.6.

All results indicate that when confronted with locally unseen data, models B1 and B2 require dynamic weighting to amplify the proportion of generalized knowledge within the shared subspace, compelling the local model to favor shared knowledge. The activation peaks and distribution patterns of models B1 and B2 after applying the weighting mechanism closely resemble those of models A1 and A2, indicating that the models attend to similar knowledge when generating outputs. Without the

weighting mechanism, models B1 and B2 do not prioritize generalized knowledge during inference. Fig.6c provides a simplified illustration of how the weighting mechanism guides local models to amplify either generalized or personalized knowledge.

We also present the response results for one sample from each task under different models, as shown in samples 1-13. Specifically, A-w/o denotes responses without dynamic weighting, A-G denotes responses dynamically weighted under a local model not specific to that task, A-P denotes responses dynamically weighted under the local model specific to that task, and A-T represents the actual standard response. The results similarly validate the effectiveness of dynamic weighted reasoning.

In summary, the weighting mechanism we designed is crucial during the inference stage and offers significant advantages. Analyzing the reasons, for the local data, increasing the proportion of parameters in the personalization decoupling space allows the model to prioritize personalized knowledge, thereby improving response quality for local tasks. Conversely, prioritizing generalized knowledge enhances the quality of responses for tasks not encountered locally. Our approach employs OOD oriented weighting. Compared to cosine similarity methods based on encoded embeddings, it enables precise determination of whether user data constitutes an OOD case. This enables more accurate parameter adjustments tailored to the user's specific data.

---

**Sample**

*Coreference task*
**Q:** Sean is better than Watson at anaphora resolution because he can naturally understand natural language. Tell me who he is.
OPTIONS: - Sean, - Watson
**A-w/o:** Watson.
**A-P:** Sean.
**A-G:** Sean.
**A-T:** Sean.

---

**Sample**

*Entailment task*
**Q:** If "A woman in a white shirt inspects a black garment while sitting in a chair at a festival.", can we conclude "A woman drinks a beer."?
OPTIONS: - yes, - it is not possible to tell, - no.
**A-w/o:** it is not possible to tell.
**A-P:** no.
**A-G:** no.
**A-T:** no.

---

**Sample**

*Linguistic acceptability task*
**Q:** Test sentence: "It is painting by Klee or drawing by Miro that the museum displays no." Is this test sentence a correct grammatical English sentence?
OPTIONS: - unacceptable, - acceptable.
**A-w/o:** acceptable.
**A-P:** unacceptable.
**A-G:** unacceptable.
**A-T:** unacceptable.

> **Sample**
>
> *Paraphrase task*
> **Q:** Are the following two questions the same? Can I use Jio 4G sim in a 3G phone? Can I use Jio sim in 3G smartphone?
> OPTIONS: - no, - yes.
> **A-w/o:** no.
> **A-P:** yes.
> **A-G:** yes.
> **A-T:** yes.

> **Sample**
>
> *Question classification task*
> **Q:** What is solar wind ?
> What kind of thing would answer this question?
> OPTIONS: - description, - entity, - abbreviation, - human, - numeric, - location.
> **A-w/o:** entity.
> **A-P:** description.
> **A-G:** description.
> **A-T:** description.

> **Sample**
>
> *Structure to text task*
> **Q:** Data: Ahmad Kadhim Assad, birthPlace, Iraq
> Sentence about the following data:
> **A-w/o:** Ahmad Kadimah is from Iraq. He was born in Baghdad.
> **A-P:** Ahmad Kadhim Assad was born in Iraq.
> **A-G:** Ahmad Kadhim Assad was born in Iraq.
> **A-T:** Ahmad Kadhim Assad's birthplace is Iraq.

> **Sample**
>
> *Text formatting task*
> **Q:** Generate a correctly punctuated version of the following text: See more about what we can do for you here on the MEGA FLEX website
> **A-w/o:** See more about what we can do for you here on the MEGA FLEX website
> **A-P:** See more about what we can do for you here on the MEGA FLEX website.
> **A-G:** See more about what we can do for you here on the MEGA FLEX website.
> **A-T:** See more about what we can do for you here on the MEGA FLEX website.

> **Sample**
>
> *Word disambiguation task*
> **Q:** 'Smog' is a blend of 'smoke' and 'fog'. Their music has been described as a blend of jazz and heavy metal. Does the word "blend" mean the same thing in the above two sentences?
> OPTIONS: - different meanings, - the same meaning.
> **A-w/o:** different meanings.
> **A-P:** the same meaning.
> **A-G:** the same meaning.
> **A-T:** the same meaning.

Sample

*Common sense task*
**Q:** Write the next sentence in this paragraph:
A man is seen speaking to the camera and begins polishing a black shoe with a brush. he
OPTIONS: - then holds up the shoe and starts waxing it with a shoe shaver. - then shows the shoe again and puts it back on. - then uses scraper to polish the shoe while still speaking and cutting with sharp edges. - points around the shoe and flips the shoe around with his hands.
**A-w/o:** then holds up the shoe and then uses scraper to polish the shoe while still speaking and cutting with sharp edges.
**A-P:** points around the shoe and flips the shoe around with his hands.
**A-G:** points around the shoe and flips the shoe around with his hands.
**A-T:** points around the shoe and flips the shoe around with his hands.

Sample

*Open domain qa task*
**Q:** What is the answer?
The heath hen, an extinct small wild fowl, was a relative of the prairie chicken. Which of the following most likely caused extinction of the heath hen?
OPTIONS: - overhunting, - stable climate, - plentiful food supply, - abundant nesting sites.
**A-w/o:** abundant nesting sites.
**A-P:** overhunting.
**A-G:** overhunting.
**A-T:** overhunting.

Sample

*Reading comprehension task*
**Q:** The return address is not required on postal mail. However, lack of a return address prevents the postal service from being able to return the item if it proves undeliverable; such as from damage, postage due, or invalid destination. Such mail may otherwise become dead letter mail.
Is it true that can you mail something with no return address?
OPTIONS: - no, - yes.
**A-w/o:** no.
**A-P:** yes.
**A-G:** yes.
**A-T:** yes.

Sample

*Sentiment task*
**Q:** The return address is not required on postal mail. However, lack of a return address prevents the postal service from being able to return the item if it proves undeliverable; such as from damage, postage due, or invalid destination. Such mail may otherwise become dead letter mail.
Is it true that can you mail something with no return address?
OPTIONS: - no, - yes.
**A-w/o:** no.
**A-P:** yes.
**A-G:** yes.
**A-T:** yes.

---

**Sample**

*Summarization task*
**Q:** a swedish un soldier in bosnia was shot and killed by a stray bullet on tuesday in an incident authorities are calling an accident , military officials in stockholm said tuesday . Can you generate a short summary of the above paragraph?
**A-w/o:** a swedish un soldier in bosnia was shot and killed by a stray bullet on tuesday in an incident authorities are calling an accident, military officials in stockholm said tuesday.
**A-P:** swedish soldier killed by stray bullet in bosnia.
**A-G:** a swedish soldier was shot and killed by a stray bullet in stockholm sweden.
**A-T:** swedish un soldier in bosnia killed by stray bullet.

---

## C.4. Convergence Analysis

We collect loss, Avg-Per and Avg-Gen during training based on the local model. For loss metrics, taking Client 1 as an example, we record the loss of its local model on the local dataset at each round, as shown in Fig.7a. The results show a clear convergence trend. For *Avg-Per* and *Avg-Gen*, we compute the average personalized and generalized performance across all clients at each round, as illustrated in Fig.7b and Fig.7c. The results indicate that as the number of training rounds, the metric data exhibits an upward trend and convergence on both Dataset 1 and Dataset 2.

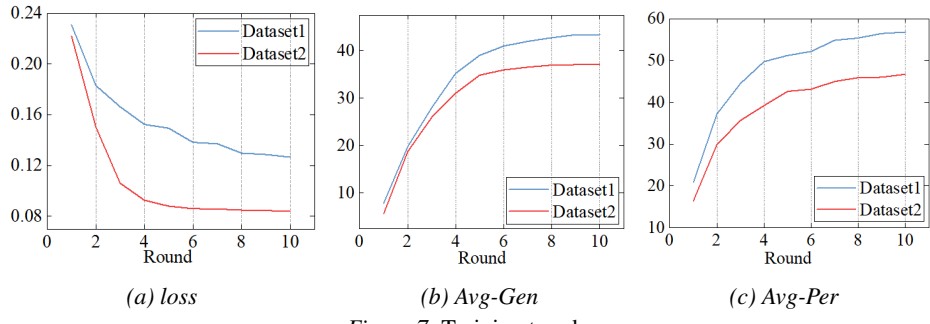

(a) loss      (b) Avg-Gen      (c) Avg-Per

*Figure 7.* Training trends.

## C.5. Overhead

Taking Dataset 1 as an example, we also evaluate the communication and local computation overhead of multiple federated algorithms in an homogeneous scenario, as shown in Table 8. Specifically, the communication overhead of our method depends on the rank $r_s$ of the shared subspace, whereas the communication overhead of the baseline methods depends on the rank $r$. Since $r_s$ is typically smaller than $r$, our method incurs lower overhead compared to the baseline methods. In terms of local computational overhead, both Het-CPFLoRA and pFedSeq involve only single-adapter fine tuning, resulting in no additional overhead and minimal computational cost. On the other hand, FedDPA and PF2LoRA incur extra overhead due to dual adapter tuning, while pFedGPT incurs additional overhead from parameter space search. Overall, our method demonstrates lower communication and computational overhead than existing approaches, providing advantages in energy consumption.

| - | Communication overhead | Computation overhead |
|---|---|---|
| PF2LoRA | $2(d \times r + k \times r)$ | 0.279 TFLOPS |
| pFedGPT | $2(d \times r + k \times r)$ | 0.277 TFLOPS |
| pFedSeq | $2(d \times r + k \times r)$ | 0.277 TFLOPS |
| FedDPA | $2(d \times r + k \times r)$ | 0.281 TFLOPS |
| Our | $2(d \times r_s + k \times r_s)$ | 0.277 TFLOPS |

*Table 8.* The communication and computation overhead on Dataset 1.

Regarding weighted inference, when LoRA matrices are merged into the base model, the weights of different LoRA

subspaces must be calculated and applied, which introduces a time delay. The average response latency per user sample is approximately $0.0512s$ on the Llama 3.2 1B model and $0.0624s$ on the 3B model. These additional delays are in the order of milliseconds and are negligible.

