# OpenReview forum: "Heterogeneous Customizable Personalized Federated  Fine-Tuning Approach for Large Language Models"
_ICML.cc/2026/Conference — ICML 2026 regular_

### Official Review · Reviewer_XM4x · 2026-03-04

**Soundness:** 3
**Presentation:** 3
**Significance:** 2
**Originality:** 3
**Overall Recommendation:** 5
**Confidence:** 3

**Summary:**

The paper presents Het-CPFLoRA, a personalized federated fine-tuning framework for LLMs that addresses data and resource heterogeneity across clients. Instead of relying on multiple LoRA adapters, the method decouples a single LoRA adapter into distinct shared and personalized parameter subspaces based on rank. To handle inference on out-of-distribution (OOD) data, the authors introduce a dynamic weighting mechanism that relies on perplexity scores to adjust the influence of the personalized versus shared parameters. The framework is evaluated on Llama 3.2 1B across two datasets encompassing various tasks, demonstrating improvements in personalization and flexibility in heterogeneous environments.

**Compliance With Llm Reviewing Policy:**

Affirmed.

**Final Justification:**

My main concerns are fully resolved.

**Key Questions For Authors:**

1. Could the authors provide theoretical motivation or empirical ablations to justify the scaling factor used in Eq. (11)?

2. How much GPU memory and computational overhead is actually reduced when varying the LoRA rank? Could the authors provide quantitative measurements for different ranks?

3. What is the practical impact of the perplexity-based OOD weighting during inference, and does it introduce noticeable latency overhead?

**Limitations:**

yes

**Strengths And Weaknesses:**

**Strengths**
- 1. The flexibility to tune the trade-off between shared and personalized knowledge purely by adjusting $r_p$ and $r_s$ offers a highly practical deployment mechanism for varying edge devices.
- 2. The LoRA subspace decoupling framework is structurally simple, avoids dual-adapter conflicts, and is easy to deploy in practice.
- 3. The experiments cover homogeneous, half-heterogeneous, and fully heterogeneous scenarios, providing relatively thorough validation.

**Weakness**
- 1. The perplexity-based OOD weighting lacks strong theoretical grounding and appears empirically motivated.
- 2. Although the paper proposes adapting to heterogeneous client resources by adjusting the LoRA rank, GPU memory consumption in practice is not determined solely by LoRA parameters. It also includes substantial costs from the base model forward/backward activations, optimizer states, attention caches, and intermediate tensors. Reducing the LoRA rank typically decreases only a fraction of the overall training and memory overhead. However, the paper does not provide quantitative analysis on how specific LoRA ranks translate to actual GPU memory usage or computational savings. This omission raises concerns about the practicality and scalability of the proposed approach under real-world edge-device heterogeneity.
- 3. While "Our" improves average generalization over FedDPA by 6.04%, this advantage largely disappears when dynamic inference is applied to both methods ("Our-auto" vs. "FedDPA-Auto"), where the gain drops to only 0.35%. This suggests that a substantial portion of the reported improvement may stem from the dynamic inference mechanism rather than the proposed federated training strategy itself.
- 4. The scaling factor in Eq. (11), $\max(r_{c,p}, r_{c,s}) / (r_{c,p} + r_{c,s})$, appears heuristic. The paper provides neither theoretical justification nor empirical ablation to support the  scaling factor design, which weakens the rigor of the dynamic weighting mechanism.

---

> ### Author Rebuttal · Authors · 2026-03-31
>
> We sincerely thank your valuable feedback. We provide clarifications to all concerns below.
>
> **W1**:
>
> This is not a design based on empirical evidence. We design this Weighting mechanism to assess whether user and local data are OOD from a semantic matching perspective. FedDPA (SOTA) is similar to ours in that it uses cosine similarity to calculate the similarity between local and user data at the token level, thereby determining the weights. However, this method has a drawback: it assigns extremely high personalization weights to user data that is similar at the token level but has significant semantic differences, which is, in fact, incorrect. We have therefore improved it by using perplexity to measure the degree of semantic matching, and subsequently calculating the weights.
>
> We conduct an ablation study in a heterogeneous setting with Llama3.2 1B and Dataset2. Using the cosine method, we get Avg-Per(46.12) and Avg-Gen(55.48), and with our method, 47.49 and 58.91. Using the FedDPA strategy to get the local and global LoRA, the cosine method gives Avg-Per(45.86) and Avg-Gen(45.32), while our method yields 46.54 and 49.17. These results demonstrate that our weighting mechanism is more effective.
>
> Furthermore, Appendix C.3 reports the changes in LoRA activation values. The activation distribution on OOD data is significantly higher after weighting than before, which demonstrates the generalization capability of our mechanism for training on unseen tasks.
>
> **W2&Q2**:
>
> We acknowledge that GPU memory consumption is not solely determined by LoRA parameters, as other factors also contribute. Studies like FlexLoRA and FLoRA show that adjusting the LoRA rank can lead to varying overheads, and propose joint fine-tuning schemes for heterogeneous edge devices. In response to your concerns, we provide GPU memory consumption data. For the Llama 3.2 1B model, memory usage is 4,717 MB, 4,824 MB, and 5,080 MB for ranks 8, 32, and 64, respectively. For the Llama 3.2 3B model, memory usage is 7,580 MB, 8,256 MB, and 8,658 MB for ranks 8, 32, and 64, respectively. As model size increases, the memory difference between ranks becomes more pronounced, with a 14% reduction in memory when the rank drops from 64 to 8.
>
> **W3**:
>
> There might be a misunderstanding.
>
> When dynamic inferencing is applied to both our method and FedDPA, the performance gap narrows. However, dynamic reasoning and our federated training strategy serve distinct purposes. The dynamic inference mechanism, similar to FedDPA’s dynamic weighting, facilitates test-time personalization by addressing distributional bias between user inputs and training data, improving performance. It provides gains across various federated strategies and can be adapted to all multi adapter methods. Notably, our baselines incorporate dynamic reasoning at test time.
>
> With regard to federated training strategies, Avg-per improved by 12.92% and Avg-gen by 6.04% under homogeneous settings, showing significant superiority. The fixed LoRA parameters per client create a performance ceiling, so Avg-gen improvement with dynamic inference is modest (0.35%) but closer to the upper bound, while Avg-per increased by 2.24%, emphasizing our method's strength in personalization while maintaining generalization.
>
> In heterogeneous settings, improvements of 16.95% and 10.99% are observed. After dynamic inference, they increase by 4.96% and 1.80%, reflecting better performance due to parameter variability and a broader ceiling.
>
> In summary, dynamic reasoning significantly improve the performance of all baselines during testing. However, due to the advantages of our training strategy, we achieve the best. The modest improvement in generalization is due to the base model's performance limits.
>
> **W4&Q1**:
>
> The scaling factors limit the personalization module's maximum score, preventing excessive weights that could hurt generalization. Inspired by FedDPA, we move beyond its experience based fixed values (0.5 and 0.3) by standardizing weights based on personalized, shared, and local ranks, prioritizing significant components. To address your concerns, we conduct ablation settings in a heterogeneous environment with five rounds of tests: Strategy A (no scaling), B (FedDPA), and C (our). The results are:
>
> A: Avg-Per = 57.01, Avg-Gen = 44.38
> B: Avg-Per = 57.22, Avg-Gen = 46.15
> C: Avg-Per = 58.91, Avg-Gen = 47.49
>
> Our design significantly improves generalization compared to A and B, confirming that the scaling factor mitigates over personalization and enhances generalization.
>
> **Q3**:
>
> The practical implication is that when LoRA matrices are merged into the base model, the weights of the different LoRA subspaces are calculated and weighted, which introduces time latency. The average response latency per user sample is approximately 0.0512s on the Llama 3.2 1B model and 0.0624s on the 3B model. These additional delays are in the order of milliseconds and are negligible.
>
> We will report them in the revision.

---

> > ### Author Rebuttal · Reviewer_XM4x · 2026-04-01
> >
> > My concerns are fully resolved, and I update my recommendation accordingly, increasing my score from 3 to 5. I look forward to seeing the final version of the paper, and it would be even stronger if the authors could provide some theoretical justification for the perplexity-based OOD weighting (W1) in the revision.

---

> > > ### Author Response · Authors · 2026-04-03
> > >
> > > Thank you very much for taking the time to provide your constructive feedback. Your comments are vital to improving the quality of my thesis, and I appreciate your recognition of my research work. We will include appropriate theoretical explanations in the revised version.

---

### Official Review · Reviewer_4sbc · 2026-03-12

**Soundness:** 2
**Presentation:** 4
**Significance:** 2
**Originality:** 2
**Overall Recommendation:** 4
**Confidence:** 4

**Summary:**

This paper addresses two key limitations of existing personalized federated LoRA fine-tuning methods: optimization conflicts in dual-adapter schemes and the inability to support heterogeneous personalization requirements across clients. By exploiting a newly discovered decoupling property of LoRA matrices, the method decomposes a single adapter into shared and personalized subspaces, eliminating the need for multiple adapters and thus resolving optimization conflicts. Each client can independently configure the rank split between the two subspaces, enabling customizable heterogeneous personalization.

**Compliance With Llm Reviewing Policy:**

Affirmed.

**Final Justification:**

The authors have addressed my concerns.

**Key Questions For Authors:**

1. Why is it believed that positional segmentation can semantically correspond to two types of knowledge? Are there any ablation experiments to verify that this segmentation method is superior to random partitioning or importance-based partitioning?
2. The experiment only set up 10 rounds of federated training. Could you provide convergence curves for more rounds?

**Limitations:**

yes

**Strengths And Weaknesses:**

strengths:
1. Both questions raised in the paper are of practical significance.
2. The method is simple. Matrix block decoupling directly reduces the multi-adaptor problem to a single-adaptor problem.

weaknesses:
1. Directly partitioning matrices B and A by column/row position lacks theoretical or experimental evidence to suggest that the "first few columns" naturally correspond to shared knowledge and the "later columns" correspond to personalized knowledge. Furthermore, the assumption that the two parts are semantically indistinguishable after initialization is unsupported.
2. The paper demonstrates trade-offs with different rank ratios, but does not provide a solution for automatically determining the optimal configuration for real-world clients.

---

> ### Author Rebuttal · Authors · 2026-03-31
>
> We sincerely thank your valuable feedback. We provide clarifications to all concerns below.
>
> **W1&Q1:**
>
> There might be a misunderstanding.
>
> It is worth clarifying that column/row positions do not inherently carry semantic meaning; however, LoRA update terms can be decomposed additively along low rank dimensions, a decomposition that allows us to assign different roles to different subspaces. In principle, this is similar to dual adapter methods such as FedDPA and PF2LoRA. These dual adapter methods consist of a local personalized adapter $\Delta W_p$ and a global adapter $\Delta W_l$, which are ultimately additively fused to form the local model parameters. This process can be viewed as the sum of two full rank LoRA matrices ($\Delta W_p+\Delta W_g \leftarrow \Delta W_l$). Building on this, we adopt a reverse perspective and propose subtractive partitioning (i.e. decoupling,  $\Delta W_l \rightarrow \Delta W_p+\Delta W_g $) of the local LoRA adapter, treating it as the additive sum of two matrices to delineate the personalised and generalization spaces.
>
> Crucially, this partitioning cannot be randomised. Section 4.1 of this paper reports the decoupling characteristics of LoRA parameters that we have discovered. Based on these characteristics and knowledge of matrix multiplication, it is clear that random partitioning would confuse the knowledge of the personalisation and shared subspaces. Furthermore, regarding the row-column partitioning of the personalisation and shared subspaces, it suffices for rows and columns to correspond.  For example, columns $x$ to $y$ of matrix $B$ correspond to rows $x$ to $y$ of matrix $A$, which are partitioned into shared or personalized subspaces. These design choices can be determined by the client, offering a high degree of flexibility.
>
> To address your concerns, we set $r_p$=4 and $r$=8, and perform ablation experiments with $X$  (the first 4 rows/columns as the shared subspace), $Y$ (the last 4 rows/columns as the shared subspace), and $Z$ (random intermediate  $x$th to $y$th rows/columns as the shared subspace, where $y$-$x$=$r_p$=4). On dataset2, Avg-Per and Avg-Gen can reach 52.27 and 45.84 ($X$), 52.88 and 45.60 ($Y$), 52.09 and 45.71 ($Z$).
>
> This demonstrates that the choice of position partitioning has little impact on the final results.
>
> **W2:**
>
> The configuration of the optimal settings is not the main focus of this paper. However, we have given this issue careful consideration and put forward suggestions for potential future optimizations to achieve the best possible settings.
>
> We recommend considering the task complexity when setting the initial values for local and personalization ranks. Task complexity dictates the parameter requirements during training. For low complexity tasks, like text formatting, a lower personalization rank suffices, as these tasks need minimal personalization. High complexity tasks, such as entailment, require more parameters for personalized knowledge and thus should have a higher personalization rank.
>
> As our experiments are conducted under randomized settings, we include ablation experiments to address your concerns and validate our recommendations. First, we categorize the tasks in Dataset 2 by complexity and assign them different personalization ranks. Training is performed under a half heterogeneous setting ($r$=8, with $r_p$ determined by task complexity) using a llama3.2 1B base model. The results are shown below.
>
> |   | Paraphrase | Common sense | Entailment | Text Formatting | Summarization | Reading Comprehension | Sentiment | OpenQA | Avg-Per | Avg-Gen |
>
> |   | $r_p$=3 | $r_p$=4 | $r_p$=5 | $r_p$=2 | $r_p$=3 |  $r_p$=4 |  $r_p$=3 | $r_p$=5 | Avg-Per | Avg-Gen |
>
> | **Our** | 39.12 | 37.88 | 54.76 | 64.13 | 27.91 | 32.11 | 62.91 | 54.17 | 47.62 | 34.01 |
>
> | **Our-auto** | 54.48 | 40.26 | 52.39 | 68.92 | 30.01 | 57.78 | 75.25 | 66.84 | 56.74 | 47.49 |
>
> The results show that, compared to random initialization (Table.2 in the paper), this approach improves personalization by 1.85% and generalization by 1.68%. After applying dynamic reasoning, these improvements rise to 2.03% and 1.36%, respectively. Personalization gains are more significant than generalization, highlighting the importance of task complexity in determining the optimal personalization rank.
>
> However, task complexity alone is insufficient. Factors such as local computational resources, data quality, and data volume also impact local model performance. In future work, we aim to integrate these factors to optimize personalization settings and enable dynamic, adaptive training under resource constraints.
>
>
> **Q2:**
>
> We extend the training to 50 iterations and collect data on loss and performance metrics; the visualize results can be found at https://anonymous.4open.science/r/Het-CPFLoRA-loss-FB57. The results demonstrate stable convergence.
>
> We will include these in the revised version. This will provide ideas and directions for future research.

---

> > ### Author Rebuttal · Reviewer_4sbc · 2026-04-01
> >
> > W1/Q1: The authors argue that random partitioning would "confuse" shared and personalized knowledge, yet the ablation (X/Y/Z) shows random intermediate positions achieve comparable results. This contradicts their own justification. A clearer explanation is needed.
> > W2: The task-complexity heuristic is a reasonable starting point but remains manual and incomplete by the authors' own admission.

---

> > > ### Author Response · Authors · 2026-04-02
> > >
> > > Dear Reviewer 4sbc, thank you very much for your careful review of our paper and thoughtful comments. We hope the following responses can help clarify potential misunderstandings and alleviate your concerns.
> > >
> > > **W1&Q1:**
> > >
> > > You may have misunderstood something.
> > >
> > > We understand that by ‘random partitioning’ you mean a fully random scheme. Our supplementary experiments are conducted under semi-random conditions. The fully random scheme violates LoRA’s decoupling property, as it conflates personalized knowledge with generalized knowledge in full rank computations. The semi-random scheme, on the other hand, offers greater flexibility without compromising LoRA’s decoupling property. Thus, random partitioning is conditional. Below, I shall explain in detail the difference between fully random and semi-random schemes.
> > >
> > > Let matrix $B \in \mathbf{R^{d \times r}}$ and $A \in \mathbf{R^{r \times k}}$, where the personalization rank is $r_p$, $r$ is the rank of the LoRA matrixs, and $d$ and $k$ are the dimensions of the matrixs.
> > >
> > > ***1.*** Fully randomized partitioning scheme. Not feasible. Not feasible!
> > >
> > > For matrix $B$, select $r_p$ columns at random, for example, columns $i$ to $j$, where $j − i = r_p-1$.
> > > For matrix $A$, select $r_p$ rows at random, for example, rows $m$ to $n$, where $n − m = r_p-1$.
> > >
> > > Here, $\mathbf{i \neq m}$ and $\mathbf{j \neq n}$.
> > >
> > > Let me give you an example, $r=4, d=k=3, r_p=2$. $i=3, j=4$, but $m=2, n=3$:
> > >
> > > $B=\begin{bmatrix}
> > >  b_{11}, b_{12}, \underline{b_{13}},\underline{b_{14}} \\\\
> > > b_{21}, b_{22},  \underline{b_{23}},\underline{b_{24}}\\\\
> > > b_{31}, b_{32}, \underline{b_{33}},\underline{b_{34}}
> > > \end{bmatrix}$
> > >
> > > $A=\begin{bmatrix}
> > > a_{11}, a_{12}, a_{13}\\\\
> > > \underline{a_{21}}, \underline{a_{22}}, \underline{a_{23}} \\\\
> > > \underline{a_{31}}, \underline{a_{32}}, \underline{a_{33}}\\\\
> > > a_{41}, a_{42}, a_{43}
> > > \end{bmatrix}$
> > >
> > > ***2.*** Semi-randomized partitioning scheme. Feasible! (Execution conditions of the supplementary experiments.)
> > >
> > > For matrix $B$, select $r_p$ columns at random, for example, columns $i$ to $j$, where $j − i = r_p-1$.
> > > For matrix $A$, select $r_p$ rows at random, for example, rows $m$ to $n$, where $n − m = r_p-1$.
> > >
> > > Here, $\mathbf{i = m}$ and $\mathbf{j = n}$.
> > >
> > > Let me give you an example, $r=4, d=k=3, r_p=2$. $i=m=2, j=n=3$:
> > >
> > > $B=\begin{bmatrix}
> > >  b_{11}, \underline{b_{12}}, \underline{b_{13}},b_{14} \\\\
> > > b_{21}, \underline{b_{22}},  \underline{b_{23}},b_{24}\\\\
> > > b_{31}, \underline{b_{32}}, \underline{b_{33}},b_{34}
> > > \end{bmatrix}$
> > >
> > > $A=\begin{bmatrix}
> > > a_{11}, a_{12}, a_{13}\\\\
> > > \underline{a_{21}}, \underline{a_{22}}, \underline{a_{23}} \\\\
> > > \underline{a_{31}}, \underline{a_{32}}, \underline{a_{33}}\\\\
> > > a_{41}, a_{42}, a_{43}
> > > \end{bmatrix}$
> > >
> > > The underlined text indicates the personalized space.
> > >
> > > Our supplementary experiments are not conducted under the fully randomized conditions of Scheme 1, but rather under the semi-randomised conditions of Scheme 2.
> > >
> > > **W2:**
> > >
> > > Optimized allocation is the focus of our future research. To address your concerns, we reviewed the literature and chose ‘action scores’ [1] (cumulative training loss) to adaptively set the difficulty of local tasks, assigning $r_p$.
> > >
> > > Firstly, $r$ is set to 8, and the initial $r_p$ is set to 4. During the first training round, the cumulative loss value for each local task is calculated and uploaded to the server as an ‘action score’. The server then sorts all ‘action scores’ and classifies them into four tiers: difficult, relatively difficult, moderate and easy. Subsequently, it adaptively assigns $r_p$ to 5, 4, 3 and 2 respectively, and notifies the client. Finally, the client uses the adjusted $r_p$ for learning. The results are shown in the table below.
> > >
> > > | | Paraphrase | Common sense | Entailment | Text Formatting | Summarization | Reading Comprehension | Sentiment | OpenQA | Avg-Per | Avg-Gen |
> > >
> > > | | $r_p$=2 | $r_p$=3 | $r_p$=5 | $r_p$=3 | $r_p$=2 | $r_p$=4 | $r_p$=4 | $r_p$=5 | Avg-Per | Avg-Gen |
> > >
> > > | Our | 39.87 | 37.82 | 54.91 | 64.13 | 28.33 | 32.74 | 62.38 | 55.88 | 48.01 | 35.22 |
> > >
> > > | Our-auto | 55.02 | 40.11 | 52.45 | 69.45 | 30.25 | 57.96 | 76.92 | 67.23 | 57.54 | 48.07 |
> > >
> > > The results indicate that it also performs exceptionally well in terms of personalization and generalization, whilst maintaining a high degree of automation.
> > >
> > > Considering complexity alone is insufficient; in the future, we will account for factors such as local resources, requirements, and data volume to enable automatic ranking and determine optimal partitioning points.
> > >
> > > [1] Cook R A, Lalor J P. No simple answer to data complexity: An examination of instance-level complexity metrics for classification tasks[C]//Proceedings of the 2025 Conference of the Nations of the Americas Chapter of the Association for Computational Linguistics. 2025: 2553-2573.
> > >
> > > In the revised edition, we'll cover optimal allocation and explain random partitioning.
> > >
> > > We would like the reviewers to reconsider our score. Thank you once again for your feedback.

---

### Official Review · Reviewer_nRAX · 2026-03-12

**Soundness:** 3
**Presentation:** 3
**Significance:** 3
**Originality:** 3
**Overall Recommendation:** 4
**Confidence:** 3

**Summary:**

The paper proposes Het-CPFLORA, a personalized federated learning framework for LLM fine-tuning that addresses both heterogeneous client resources and diverse personalization requirements. The method leverages a LoRA matrix decoupling strategy to separate shared and personalized subspaces, enabling flexible local ranks while mitigating optimization conflicts. It further introduces rank-aware aggregation and a perplexity-based OOD dynamic inference mechanism to improve robustness and personalization.

**Compliance With Llm Reviewing Policy:**

Affirmed.

**Final Justification:**

The authors have addressed my concerns, thus i keep my positive score.

**Key Questions For Authors:**

1. How should a client optimally choose the initial rank $r_c$ and the personalized split $r_{c,p}$? Is there a heuristic based on local data volume or task complexity that you recommend?
2. The rank-aware aggregation weight $\alpha_c^t$ incorporates $r_{c,s}$. In a scenario where a client has high data volume but chooses a very low shared rank (to prioritize privacy/personalization), how does this affect the global model's performance on that client's specific task domain?
3. How does the method perform on different model architectures, such as Qwen? Additionally, how does its performance vary under different LoRA ranks?
4. The paper does not provide a discussion of the limitations or potential weaknesses of the proposed method, which would help readers better understand its scope and applicability.
5. Does “Test-Time Personalization / Generalization” involve test-time learning?

**Limitations:**

Please see the question.

**Strengths And Weaknesses:**

The paper is well written and clearly organized, making the proposed ideas easy to follow. The experimental evaluation is comprehensive, with thorough comparisons and analyses that support the effectiveness of the method. The authors provide theoretical analysis to support the proposed method.

---

> ### Author Rebuttal · Authors · 2026-03-31
>
> We sincerely thank your valuable feedback. We provide clarifications to all concerns below.
>
> **Q1:**
>
> We recommend considering the task complexity when setting the initial values for local and personalization ranks. Task complexity dictates the parameter requirements during training. For low complexity tasks, like text formatting, a lower personalization rank suffices, as these tasks need minimal personalization. High complexity tasks, such as entailment, require more parameters for personalized knowledge and thus should have a higher personalization rank.
>
> As our experiments are conducted under randomized settings, we include ablation experiments to address your concerns and validate our recommendations. First, we categorize the tasks in Dataset 2 by complexity and assign them different personalization ranks. Training is performed under a half heterogeneous setting ($r$=8, with $r_p$ determined by task complexity) using a llama3.2 1B base model. The results are shown below.
>
> |   | Paraphrase | Common sense | Entailment | Text Formatting | Summarization | Reading Comprehension | Sentiment | OpenQA | Avg-Per | Avg-Gen |
>
> |   | $r_p$=3 | $r_p$=4 | $r_p$=5 | $r_p$=2 | $r_p$=3 |  $r_p$=4 |  $r_p$=3 | $r_p$=5 | Avg-Per | Avg-Gen |
>
> | **Our** | 39.12 | 37.88 | 54.76 | 64.13 | 27.91 | 32.11 | 62.91 | 54.17 | 47.62 | 34.01 |
>
> | **Our-auto** | 54.48 | 40.26 | 52.39 | 68.92 | 30.01 | 57.78 | 75.25 | 66.84 | 56.74 | 47.49 |
>
> The results show that, compared to random initialization (Table.2 in the paper), this approach improves personalization by 1.85% and generalization by 1.68%. After applying dynamic reasoning, these improvements rise to 2.03% and 1.36%, respectively. Personalization gains are more significant than generalization, highlighting the importance of task complexity in determining the optimal personalization rank.
>
> However, task complexity alone is insufficient. Factors such as local computational resources, data quality, and data volume also impact local model performance. In future work, we aim to integrate these factors to optimize personalization settings and enable dynamic, adaptive training under resource constraints.
>
> **Q2:**
>
> This is an interesting question. In extreme cases, a large volume of data with a low $r_{c,s}$ can lead to disproportionately high $\alpha_c^t$, causing the global model to focus on the shared knowledge from that client. Given the client’s abundant local data, the shared subspace parameters contain much task specific knowledge. As a result, the global model performs slightly worse than the local model on that task, but the difference is not significant.
> We test with eight clients: the text formatting client has1,000 data points, while others have 200. $r_{c,s}$ is 1 for the text formatting client and 4 for the others, with $r_{c}$ is 8. The global model achieve a score of 66.99, about 4% lower than the local model’s score of 71.43.
>
> **Q3:**
>
> We conduct scalability experiments with the Qwen 2.5 1.5B model and Dataset 2. In a heterogeneous environment, our method improves Avg-per by 6.64% and Avg-gen by 4.05%. After integrating dynamic weighting, personalization and generalization improve by 4.33% and 3.06%, respectively. Overall, the method continues to demonstrate robustness and superiority.
>
> Using the text formatting task as an example, we present the changes in local model performance with $r_p$={1, 2, 3, 4}, and $r$=8. The generalization and personalization capabilities are 49, 22, 55.15 ($r_p$=1); 48, 73, 59.01 ($r_p$=2); 47, 96, 65.92 ($r_p$=3); 46, 31, 67.12 ($r_p$=4). As our paper shows, increasing the personalization rank improves the client's performance on local tasks, but reduces its overall generalization across other tasks.
>
> **Q4**
>
> We propose the first customizable personalized federated LoRA fine-tuning approach, leveraging LoRA's decoupling characteristics. This method supports a wide range of federated learning environments, including homogeneous and heterogeneous resources, and diverse personalization needs, demonstrating strong universality. However, we acknowledge the following limitations:
>
> 1. Fixed Heterogeneous Setup: The method assumes static personalization requirements, lacking the ability to adaptively configure LoRA and optimize it when local resources and needs change.
> 2. Security Environment: The method assumes a trusted execution environment, where local data and model parameters are securely accessible with no adversarial attacks.
>
> In future work, we aim to implement adaptive adjustments for personalized and generalized knowledge under resource constraints, focusing on optimizing personalized LoRA setups.
>
> **Q5：**
>
> Our test-time personalization/generalization does not involve learning during testing. It refers to evaluating the model on unseen data, including local client-specific tasks and generalization tasks from other clients, to assess its personalization and generalization capabilities.
>
> We will report these in the revised version.

---

> > ### Author Rebuttal · Reviewer_nRAX · 2026-04-03
> >
> > Thank you for your response. I will keep my score.

---

> > > ### Author Response · Authors · 2026-04-03
> > >
> > > Thank you very much for taking the time to provide your constructive feedback. Your comments are vital to improving the quality of my thesis, and I appreciate your recognition of my research work.

---

### Official Review · Reviewer_3BrF · 2026-03-13

**Soundness:** 3
**Presentation:** 3
**Significance:** 3
**Originality:** 3
**Overall Recommendation:** 4
**Confidence:** 4

**Summary:**

This paper presents a method called Het-CPFLoRA. Specifically, it decouples a single LoRA adapter into shared and private subspaces, then aggregates the shared part across clients with SVD-based compression. Empirically, this paper evaluates the method intensively and demonstrates good performance over baselines.

**Compliance With Llm Reviewing Policy:**

Affirmed.

**Key Questions For Authors:**

See Weaknesses.

**Limitations:**

See weaknesses.

**Strengths And Weaknesses:**

Strengths:
1. This paper tackles a meaningful challenge, the conflicts of local and global personalization optimization updates.
2. To the best of the reviewer's knowledge, the proposed methods are new and novel, by splitting the subspace of LoRA.
3. Empirically, this paper achieves good improvements over baselines. This paper evaluates the paper extensively.

Weaknesses:
1. How to find the sweet point for the subspace rank split is good question. Although this paper proposes the ablation on this, however, there is significant improvements to the averaged score if we increase the rank. What is the best split for this?

Minor:
1. It is more common to place the table captions at the top of table.

---

> ### Author Rebuttal · Authors · 2026-03-31
>
> We sincerely thank your valuable feedback. We provide clarifications to all concerns below.
>
> **W1:**
> Blindly increasing the rank may lead to the model becoming either over-fit or over-generalized. Finding the optimal point remains a challenge. After careful consideration, we have put forward some recommendations.
>
> The complexity of local tasks may be a factor in determining the optimal configuration. Task complexity dictates the parameter requirements during training. For low complexity tasks, like text formatting, a lower personalization rank suffices, as these tasks need minimal personalization. High complexity tasks, such as entailment, require more parameters for personalized knowledge and thus should have a higher personalization rank.
>
> As our experiments are conducted under randomized settings, we include ablation experiments to address your concerns and validate our recommendations. First, we categorize the tasks in Dataset 2 by complexity and assign them different personalization ranks. Training is performed under a half heterogeneous setting ($r$=8, with $r_p$ determined by task complexity) using a llama3.2 1B base model. The results are shown below.
>
> |   | Paraphrase | Common sense | Entailment | Text Formatting | Summarization | Reading Comprehension | Sentiment | OpenQA | Avg-Per | Avg-Gen |
>
> |   | $r_p$=3 | $r_p$=4 | $r_p$=5 | $r_p$=2 | $r_p$=3 |  $r_p$=4 |  $r_p$=3 | $r_p$=5 | Avg-Per | Avg-Gen |
>
> | **Our** | 39.12 | 37.88 | 54.76 | 64.13 | 27.91 | 32.11 | 62.91 | 54.17 | 47.62 | 34.01 |
>
> | **Our-auto** | 54.48 | 40.26 | 52.39 | 68.92 | 30.01 | 57.78 | 75.25 | 66.84 | 56.74 | 47.49 |
>
> The results show that, compared to random initialization (Table.2 in the paper), this approach improves personalization by 1.85% and generalization by 1.68%. After applying dynamic reasoning, these improvements rise to 2.03% and 1.36%, respectively. Personalization gains are more significant than generalization, highlighting the importance of task complexity in determining the optimal personalization rank.
>
> However, task complexity alone is insufficient. Factors such as local computational resources, data quality, and data volume also impact local model performance. In future work, we aim to integrate these factors to optimize personalization settings and enable dynamic, adaptive training under resource constraints.
>
> **M1:**
>
> We will modify these tables in the revised version.
>
> We will report these in the revised version.

---

> > ### Author Rebuttal · Reviewer_3BrF · 2026-04-02
> >
> > Thanks for clarifying! I will maintain my positive score.

---

> > > ### Author Response · Authors · 2026-04-03
> > >
> > > Thank you very much for taking the time to provide your constructive feedback. Your comments are vital to improving the quality of my thesis, and I appreciate your recognition of my research work.

---

### Decision · Program_Chairs · 2026-04-30

**Decision:**

Accept (regular)

**Comment:**

This paper proposes Het-CPFLoRA, a personalized federated fine-tuning framework for LLMs that addresses client heterogeneity in both resources and personalization needs. The core idea involves decoupling a single LoRA adapter into shared and personalized subspaces via rank partitioning, thereby avoiding the optimization conflicts inherent in dual-adapter approaches. The method further incorporates an OOD-aware dynamic weighting mechanism for inference.

The submission received four reviews with scores ranging from 4 to 5. Reviewers generally acknowledged the practical significance of addressing heterogeneous personalization requirements and the simplicity of the proposed decoupling strategy. Empirical results demonstrated improvements over state-of-the-art baselines across various heterogeneity settings.

During the rebuttal phase, the authors provided additional ablation studies comparing different partitioning strategies (positional vs. semi-random), demonstrating that the specific position matters less than maintaining the structural correspondence between matrices A and
B. They also introduced a heuristic based on task complexity (and subsequently, cumulative loss) to guide rank allocation, showing measurable gains. Furthermore, the authors supplied quantitative data on GPU memory consumption across different ranks and model sizes, as well as latency measurements for the inference mechanism, which were found to be negligible.

After the rebuttal, all reviewers indicated that their main concerns were adequately addressed. While the theoretical grounding for the perplexity-based weighting could be strengthened, the empirical validation is comprehensive, and the proposed method offers a flexible, deployable solution for heterogeneous federated learning scenarios. Given the consensus among reviewers, I recommend acceptance.